# Confidence-aware Denoised Fine-tuning of Off-the-shelf Models for Certified Robustness

**Suhyeok Jang***        *jasuhe900@kaist.ac.kr*
*Korea Advanced Institute of Science & Technology (KAIST)*

**Seojin Kim***        *osikjs@kaist.ac.kr*
*Korea Advanced Institute of Science & Technology (KAIST)*

**Jinwoo Shin**        *jinwoos@kaist.ac.kr*
*Korea Advanced Institute of Science & Technology (KAIST)*

**Jongheon Jeong**        *jonghj@korea.ac.kr*
*Korea University*

*\* The authors contributed equally.*

**Reviewed on OpenReview:** *https://openreview.net/forum?id=99GovbuMcP*

## Abstract

The remarkable advances in deep learning have led to the emergence of many off-the-shelf classifiers, *e.g.*, large pre-trained models. However, since they are typically trained on clean data, they remain vulnerable to adversarial attacks. Despite this vulnerability, their superior performance and transferability make off-the-shelf classifiers still valuable in practice, demanding further work to provide adversarial robustness for them in a *post-hoc* manner. A recently proposed method, *denoised smoothing*, leverages a denoiser model in front of the classifier to obtain *provable robustness* without additional training. However, the denoiser often creates *hallucination*, *i.e.*, images that have lost the semantics of their originally assigned class, leading to a drop in robustness. Furthermore, its noise-and-denoise procedure introduces a significant distribution shift from the original distribution, causing the denoised smoothing framework to achieve sub-optimal robustness. In this paper, we introduce *Fine-Tuning with Confidence-Aware Denoised Image Selection (FT-CADIS)*, a novel fine-tuning scheme to enhance the certified robustness of off-the-shelf classifiers. FT-CADIS is inspired by the observation that the *confidence* of off-the-shelf classifiers can effectively identify hallucinated images during denoised smoothing. Based on this, we develop a confidence-aware training objective to handle such hallucinated images and improve the stability of fine-tuning from denoised images. In this way, the classifier can be fine-tuned using only images that are beneficial for adversarial robustness. We also find that such a fine-tuning can be done by merely updating a small fraction (*i.e.*, 1%) of parameters of the classifier. Extensive experiments demonstrate that FT-CADIS has established the state-of-the-art certified robustness among denoised smoothing methods across all $\ell_2$-adversary radius in a variety of benchmarks, such as CIFAR-10 and ImageNet.

## 1 Introduction

Despite the recent advancements in modern deep neural networks in various computer vision tasks (Radford et al., 2021; Rombach et al., 2022; Kirillov et al., 2023), they still suffer from the presence of *adversarial examples* (Szegedy et al., 2013) *i.e.*, a non-recognizable perturbation (for humans) of an image often fools the image classifiers to flip the output class (Goodfellow et al., 2014). Such adversarial examples can

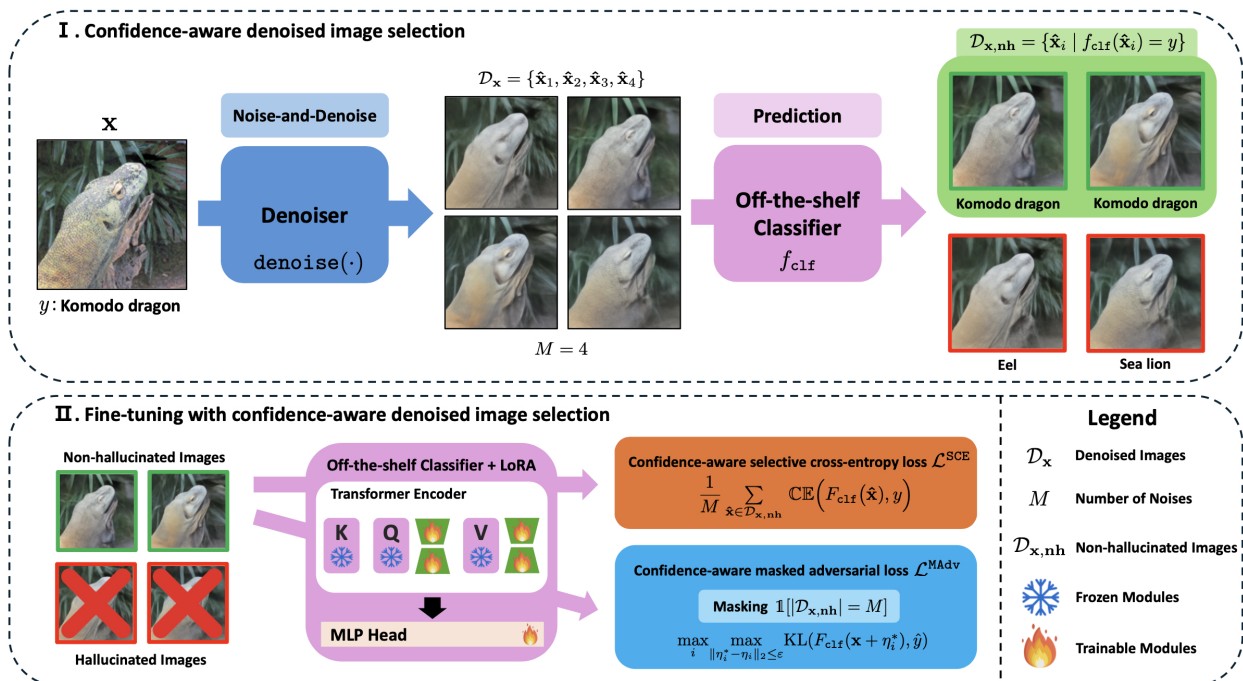

Figure 1: Overview of FT-CADIS framework. (1) Confidence-aware denoised image selection: for a given clean image, we create denoised images and find non-hallucinated images. (2) Fine-tuning with confidence-aware denoised image selection: we propose fine-tuning objectives to improve both generalizability and robustness of the smoothed classifier based on selected non-hallucinated images.

be artificially crafted with malicious intent, *i.e.*, *adversarial attacks*, which pose a significant threat to the practical deployment of deep neural networks. To alleviate this issue, various approaches have been proposed to develop *robust* neural networks, such as adversarial training (Madry et al., 2018; Wang et al., 2019) and certified defenses (Wong & Kolter, 2018; Cohen et al., 2019; Li et al., 2023).

Among these efforts, *randomized smoothing* (Lecuyer et al., 2019; Cohen et al., 2019) has gained much attention as a framework to build robust classifiers. This is due to its superior provable guarantee of the non-existence of adversarial examples, *i.e.*, certified robustness (Wong & Kolter, 2018; Xiao et al., 2018), under any perturbations confined in a $\ell_2$-norm. Specifically, it builds a *smoothed classifier* through taking a majority vote from a *base classifier*, *e.g.*, a neural network, under Gaussian perturbations of the given input image. However, it has been practically challenging to scale the model due to a critical drawback: the base classifier should be specifically trained on noise-augmented data (Lecuyer et al., 2019; Cohen et al., 2019).

Recently, Lee (2021); Carlini et al. (2023) have introduced *denoised smoothing* which utilizes pre-trained off-the-shelf classifiers within the randomized smoothing framework. Rather than directly predicting the label of a noise-augmented image, it first feeds the perturbed image into a *denoiser*, *e.g.*, a diffusion model, and then obtains the predicted label of the denoised image using off-the-shelf pre-trained classifiers that have been trained on clean images. Intriguingly, denoised smoothing with recently developed diffusion models and pre-trained classifiers, *e.g.*, guided diffusion (Dhariwal & Nichol, 2021) and BEiT (Bao et al., 2022), shows its superior scalability with comparable certified robustness in $\ell_2$-adversary to the current state-of-the-art methods (Horváth et al., 2022b; Jeong et al., 2023).

On the other hand, denoised smoothing also exhibits clear limitations. Firstly, denoised images do not follow the standard pre-training distribution of the classifiers, which results in a limited robustness of the denoised smoothing framework. Secondly, fine-tuning the pre-trained classifiers with the denoised images also yields sub-optimal classifiers due to the *hallucinated* images (Carlini et al., 2023), *i.e.*, the diffusion denoiser tends to generate image semantics from an incorrect class rather than the originally assigned class (see Figure 2a).

Consequently, denoised smoothing with such classifiers leads to a drop of the certified accuracy, especially in the large $\ell_2$-radius regime, *i.e.*, high Gaussian variance (see Table 1b).

**Contribution.** In this paper, we aim to address the aforementioned issues of denoised smoothing by designing a fine-tuning objective for off-the-shelf classifiers that distinguishes between *hallucinated* images, *i.e.*, images that have lost the original semantics after denoising, and *non-hallucinated* images, *i.e.*, images that maintain the original semantics after denoising. To this end, we propose to use the "likelihood of denoised images", *i.e.*, *confidence*, of the off-the-shelf classifier with respect to the originally assigned class as a proxy for determining whether an image is hallucinated and then fine-tune the classifier with non-hallucinated images only. Consequently, we have developed a confidence-aware training objective based on the likelihood of denoised images to effectively discriminate hallucinated images (see Figure 1).

Specifically, we propose a scalable and practical framework for fine-tuning off-the-shelf classifiers, coined *Fine-Tuning with Confidence-Aware Denoised Image Selection* (FT-CADIS), which improves certified robustness under denoised smoothing. In order to achieve this, two new losses are defined: the *Confidence-aware selective cross-entropy loss* and the *Confidence-aware masked adversarial loss*. Two losses are selectively applied only to non-hallucinated images, thereby ensuring that the overall training process avoids over-optimizing hallucinated samples, *i.e.*, samples that are harmful for generalization, while maximizing the robustness of smoothed classifiers. Our particular loss design is motivated by Jeong et al. (2023), who were the first to investigate training objectives for randomized smoothing depending on sample-wise confidence information. We demonstrate that our novel definition of confidence in randomized smoothing, specifically through the ratio of non-hallucinated images from a denoiser, can dramatically stabilize the confidence-aware training, overcoming its previous limitation of severe accuracy degradation (*e.g.*, see Table 1b).

In our experiments, we have validated the effectiveness of our proposed method on standard benchmarks for certified $\ell_2$-robustness, *i.e.*, CIFAR-10 (Krizhevsky, 2009) and ImageNet (Russakovsky et al., 2015). Our results show that the proposed method significantly outperforms existing state-of-the-art denoised smoothing methods in certified robustness across all $\ell_2$-norm setups, while updating only 1% of the parameters of off-the-shelf classifiers on ImageNet. In particular, FT-CADIS significantly improves the certified robustness in the high Gaussian variance regime, *i.e.*, high certified radius. For instance, FT-CADIS outperforms the best performing baseline, *i.e.*, diffusion denoised (Carlini et al., 2023), by $29.5\% \rightarrow 39.4\%$ at $\varepsilon = 2.0$ for ImageNet experiments.

## 2 Preliminaries

**Adversarial robustness and randomized smoothing.** We assume a labeled dataset $D = \{(\mathbf{x}_i, y_i)\}_{i=1}^{n}$ sampled from $P$, where $\mathbf{x}_i \in \mathcal{X} \subset \mathbb{R}^d$ and $y_i \in \mathcal{Y} := \{1, ..., K\}$, and aim to develop a classifier $f : \mathcal{X} \rightarrow \mathcal{Y}$ which correctly classifies a given input $\mathbf{x}$ into the corresponding label among $K$ classes, *i.e.*, $f(\mathbf{x}_i) = y_i$.

*Adversarial robustness* refers to the *worst-case* behavior of $f$; given a sample $\mathbf{x} \in \mathcal{X}$ and the corresponding label $y \in \mathcal{Y}$, it requires $f$ to produce a consistent output under any perturbation $\delta \in \mathbb{R}^d$ which preserves the original semantic of $\mathbf{x}$. Here, $\delta$ is commonly assumed to be restricted in some $\ell_2$-norm in $\mathbb{R}^d$, *i.e.*, $\|\delta\|_2 \leq \varepsilon$ for some positive $\varepsilon$. For example, Moosavi-Dezfooli et al. (2016); Carlini et al. (2019) quantify adversarial robustness as *average minimum distance* of the perturbations that cause $f$ to flip the originally assigned label $y$, defined as:

$$R(f; P) := \mathbb{E}_{(\mathbf{x}, y) \sim P} \left[ \min_{f(\mathbf{x}') \neq y} \|\mathbf{x}' - \mathbf{x}\|_2 \right] . \tag{1}$$

The primary obstacle in achieving adversarial robustness lies in the difficulty of evaluating and optimizing for it, which is typically infeasible because $f$ is usually modeled by a complex, high-dimensional neural network. *Randomized smoothing* (Cohen et al., 2019; Lecuyer et al., 2019) addresses this challenge by constructing a new robust classifier $g$ from $f$, instead of directly modeling robustness with $f$. In particular, Cohen et al. (2019) models $g$ by selecting the *most probable* output of $f$ under Gaussian noise $\mathcal{N}(0, \sigma^2 \boldsymbol{I})$, defined as:

$$g(\mathbf{x}) := \arg\max_{c \in \mathcal{Y}} \mathbb{P}_{\delta \sim \mathcal{N}(0, \sigma^2 \boldsymbol{I})}[f(\mathbf{x} + \delta) = c] . \tag{2}$$

Intriguingly, $g$ can *guarantee* the adversarial robustness around $(\mathbf{x}, y) \sim P$, *i.e.*, $R(g; \mathbf{x}, y)$ can be lower-bounded by the *certified radius* $\underline{R}(g, \mathbf{x}, y)$, where Cohen et al. (2019) have proven that such a lower-bound of certified radius is tight for $\ell_2$-adversary:

$$R(g; \mathbf{x}, y) \geq \sigma \cdot \Phi^{-1}(p_g(\mathbf{x}, y)) =: \underline{R}(g, \mathbf{x}, y), \quad \text{where} \quad p_g(\mathbf{x}, y) := \mathbb{P}_\delta[f(\mathbf{x} + \delta) = y], \tag{3}$$

provided that $g(\mathbf{x}) = y$, *i.e.*, $y$ is the most probable output of $f$ under Gaussian noise. Otherwise, we have $R(g; \mathbf{x}, y) := 0$. Here, $\Phi$ is the cumulative distribution function of the standard Gaussian distribution. We remark that higher $p_g(\mathbf{x}, y)$, *i.e.*, average accuracy of $f(\mathbf{x} + \delta)$, results in higher robustness.

**Denoised smoothing.** In randomized smoothing, it is crucial that $f$ consistently classifies perturbed images correctly. Salman et al. (2020) have proposed to define $f$ based on concatenating a Gaussian denoiser, denoted as $\texttt{denoise}(\cdot)$, with any off-the-shelf classifier $f_{\texttt{clf}}$, *i.e.*, trained with non-perturbed images, a method referred to as *denoised smoothing*:

$$f(\mathbf{x} + \delta) := f_{\texttt{clf}}(\texttt{denoise}(\mathbf{x} + \delta)) . \tag{4}$$

Denoised smoothing provides a more scalable framework for randomized smoothing. First, we only need off-the-shelf pre-trained classifiers (rather than noise-specialized classifiers), which is widely investigated and developed (Dosovitskiy et al., 2020; Bao et al., 2022; Radford et al., 2021). Second, recent advancements in *diffusion models* (Ho et al., 2020; Nichol & Dhariwal, 2021; Dhariwal & Nichol, 2021) have produced appropriate denoisers for this approach. Previous efforts (Lee, 2021; Carlini et al., 2023) have further demonstrated the potential of denoised smoothing in achieving the state-of-the-art certified robustness when combined with recently advanced pre-trained classifiers and diffusion models.

**Parameter-efficient fine-tuning.** LoRA (Hu et al., 2022) is a widely-used parameter-efficient fine-tuning method that originated from language models. It applies a low-rank constraint to approximate the update matrix at each layer of the Transformer's self-attention layer, significantly reducing the number of trainable parameters for downstream tasks. During fine-tuning, all the parameters of the original model are frozen, and the update of the layer is constrained by representing them with a low-rank decomposition. A forward pass $h = \boldsymbol{W}_0 x$ can be modified as follows:

$$h = \boldsymbol{W}_0 x + \Delta \boldsymbol{W} x = \boldsymbol{W}_0 x + \boldsymbol{B} \boldsymbol{A} x, \tag{5}$$

where $x$ and $h$ denote the input and output features of each layer, $\boldsymbol{W}_0 \in \mathbb{R}^{d \times k}$ represents the original weights of the base model $f$, while $\Delta \boldsymbol{W}$ denotes the weight change, composed of the inserted low-rank matrices $\boldsymbol{B} \in \mathbb{R}^{d \times r}$ and $\boldsymbol{A} \in \mathbb{R}^{r \times k}$.

## 3 Method

In Section 3.1, we present a description of our problem and the main idea. In Section 3.2, we provide descriptions of our selection strategy for non-hallucinated samples. In Section 3.3, we outline our overall fine-tuning framework.

### 3.1 Problem description and Overview

In this paper, we investigate *how* to effectively elaborate an off-the-shelf classifier $f_{\texttt{clf}}$ within a denoised smoothing scheme. We remark that the robustness of the smoothed classifier $g$ from denoised smoothing of $f_{\texttt{clf}}$ depends directly on the accuracy of the *denoised* images (see Eq. (3) and (4)). Therefore, one may expect that improving $f_{\texttt{clf}}$ for clean images is sufficient to improve the generalizability and robustness of $g$ (Carlini et al., 2023), assuming that the denoised images follow the pre-training distribution with clean images (Salman et al., 2020), *i.e.*, the denoised images preserve the semantics of the original clean images. However, this assumption is not true; the noise-and-denoise procedure of denoised smoothing often suffers from distribution shifts and *hallucination* issues so that the resulting denoised images have completely different semantics from the original labels (see Figure 2a).

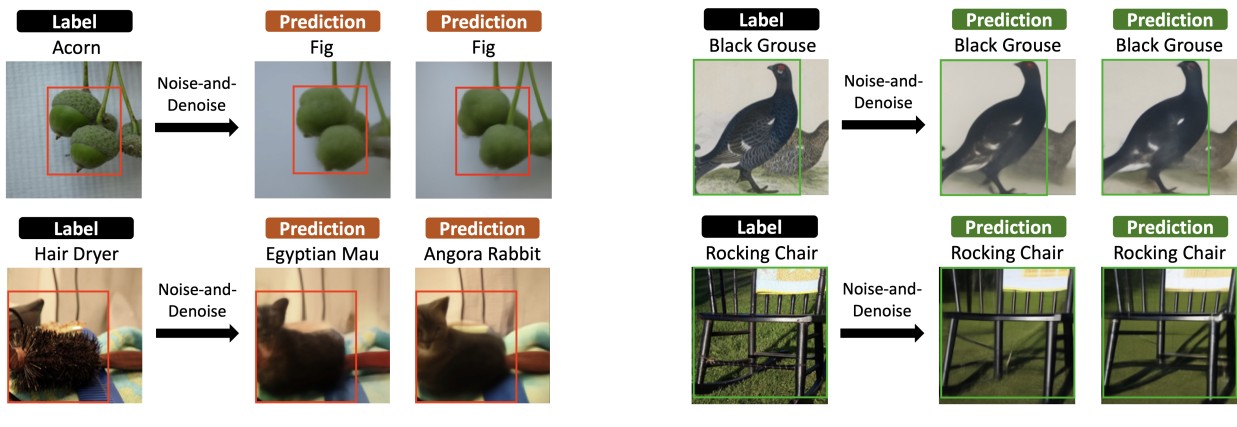

(a) Hallucinated images                    (b) Non-hallucinated images

Figure 2: Examples of denoised images for FT-CADIS on ImageNet at $\sigma = 1.00$. We visualize (a) hallucinated images and (b) non-hallucinated images after the noise-and-denoise procedure. The red/green box indicates the areas where the original semantic of the image is corrupted/preserved, respectively.

To alleviate these issues, we aim to develop a fine-tuning scheme for $f_{\texttt{clf}}$ to properly handle denoised samples. One straightforward strategy would be to fine-tune $f_{\texttt{clf}}$ by minimizing the cross-entropy loss with *all* denoised images (Carlini et al., 2023):

$$\mathcal{L}^{\text{CE}} := \frac{1}{M} \sum_{i=1}^{M} \mathbb{CE}\Big(f_{\texttt{clf}}\big(\texttt{denoise}(\mathbf{x} + \delta_i)\big), y\Big), \ \delta_i \sim \mathcal{N}(0, \sigma^2 \boldsymbol{I}), \tag{6}$$

where $\mathbb{CE}$ denotes the cross-entropy loss, and $M$ denotes the number of noises. Here, we note that this approach treats both *non-hallucinated* and *hallucinated* samples equally among the denoised samples. However, fine-tuning $f_{\texttt{clf}}$ with hallucinated samples, *i.e.*, $\texttt{denoise}(x+\delta_i)$ does not resemble the class $y$, is harmful for the generalizability since Eq. (6) forces the classifier $f_{\texttt{clf}}$ to *remember* non-$y$-like hallucinated images as $y$. Our contribution lies in resolving this issue by introducing (1) a *confidence-aware* selection strategy to distinguish between hallucinated and non-hallucinated images and (2) a fine-tuning strategy that excludes hallucinated samples from the optimization process.

### 3.2 Confidence-aware denoised image selection

We propose a confidence-aware selection strategy to identify hallucinated images and non-hallucinated images within a set of denoised images. Consider the denoised images $\mathcal{D}_{\mathbf{x}} = \{\texttt{denoise}(\mathbf{x}+\delta_1), ..., \texttt{denoise}(\mathbf{x}+\delta_M)\}$ for a given clean image $\mathbf{x}$ and the number of noises $M$. We aim to find non-hallucinated images within $\mathcal{D}_{\mathbf{x}}$ that an off-the-shelf classifier $f_{\texttt{clf}}$ classifies as the assigned label $y$, *i.e.*, $f_{\texttt{clf}}$ shows the highest confidence for $y$ among all possible classes. Conversely, if $f_{\texttt{clf}}$ classifies denoised images as a label other than $y$, we define such denoised images as hallucinated images, *i.e.*, samples that no longer preserve the core semantic of $y$. Accordingly, the set of non-hallucinated images $\mathcal{D}_{\mathbf{x},\mathbf{nh}} \in \mathcal{D}_{\mathbf{x}}$ is defined as follows:

$$\mathcal{D}_{\mathbf{x},\mathbf{nh}} = \{\hat{\mathbf{x}}|f_{\texttt{clf}}(\texttt{denoise}(\mathbf{x} + \delta_i)) = y, \ i \in [1, ..., M]\} \ . \tag{7}$$

We remark that the off-the-shelf classifier $f_{\texttt{clf}}$ is pre-trained with clean images, rather than denoised images. Thus, at the beginning of the fine-tuning, $f_{\texttt{clf}}$ often fails to correctly assign $\mathcal{D}_{\mathbf{x},\mathbf{nh}}$ due to the distribution shift from clean images to denoised images. Thus, we update $\mathcal{D}_{\mathbf{x},\mathbf{nh}}$ at each training iteration using Eq. (7) for a more accurate assignment of non-hallucinated images.

### 3.3 Fine-tuning with confidence-aware denoised image selection

Our main goal is to improve both the generalizability and the robustness of the smoothed classifier $g$, through the fine-tuning of the off-the-shelf classifier $f_{\texttt{clf}}$ based on our confidence-aware denoised image selection in

Section 3.2. To this end, we propose two fine-tuning objectives for an off-the-shelf classifier $f_{\texttt{clf}}$, *viz.*, Confidence-aware selective cross-entropy loss and Confidence-aware masked adversarial loss, to maximize the generalizability and robustness of the corresponding smoothed classifier $g$, respectively.

**Confidence-aware selective cross-entropy loss.** We first aim to improve the generalizability of the smoothed classifier $g$, *i.e.*, the average certified accuracy of $g$. Specifically, we propose to optimize $f_{\texttt{clf}}$ with non-hallucinated images $\mathcal{D}_{\mathbf{x},\mathbf{nh}}$:

$$\mathcal{L}^{\texttt{SCE}} := \frac{1}{M} \sum_{\hat{\mathbf{x}} \in \mathcal{D}_{\mathbf{x},\mathbf{nh}}} \mathbb{CE}\Big(f_{\texttt{clf}}(\hat{\mathbf{x}}),\, y\Big) . \tag{8}$$

In other words, we optimize our classifier with the non-hallucinated images, while the hallucinated images are excluded from our training objective. This prevents the drop in accuracy of $f_{\texttt{clf}}$ caused by being forced to remember wrong semantics not relevant to the assigned class $y$. It also allows for $f_{\texttt{clf}}$ to properly learn the distribution of *denoised* images, which is largely different from its pre-training distribution with clean images.

Here, we find that training with the objective in Eq. (8) slows down the overall training procedure since $\mathcal{D}_{\mathbf{x},\mathbf{nh}} = \emptyset$ sometimes occurs at the start of training. This is mainly due to the distribution shift from the pre-training clean image distribution to the denoised images, *i.e.*, $f_{\texttt{clf}}$ fails to classify denoised images due to insufficient exposure to denoised images. To resolve this *cold-start* problem, we add the most-$y$-like denoised image, *i.e.*, a denoised image with the largest logit for $y$, to $\mathcal{D}_{\mathbf{x},\mathbf{nh}}$ when it is empty.

**Confidence-aware masked adversarial loss.** We also propose a simple strategy to further improve the robustness of the smoothed classifier $g$, *i.e.*, the certified accuracy of $g$ at large $\ell_2$-norm radius. Specifically, we apply the concept of *adversarial training* (Madry et al., 2018; Zhang et al., 2019a; Wang et al., 2019; Salman et al., 2019; Jeong et al., 2023) to our denoised smoothing setup; we carefully create more challenging images, and then additionally learn these images during fine-tuning. Here, the main challenge is to ensure that the adversarial images *preserve* the core semantic of the original image, thereby maintaining generalizability while improving robustness. However, as illustrated in Figure 2, some clean images are prone to be hallucinated after the noise-and-denoise procedure. Therefore, adversarial training in denoised smoothing should be carefully designed to avoid incorporating hallucinated images.

To this end, we propose to create adversarial examples based only on images that are unlikely to be hallucinated, *i.e.*, clean images $\mathbf{x}$ with $\mathcal{D}_{\mathbf{x},\mathbf{nh}} = \mathcal{D}_{\mathbf{x}}$. Specifically, we apply our adversarial loss based on a simple condition of "$\mathcal{D}_{\mathbf{x},\mathbf{nh}} = M$":

$$\mathcal{L}^{\texttt{MAdv}} := \mathbb{1}[|\mathcal{D}_{\mathbf{x},\mathbf{nh}}| = M] \cdot \max_i \max_{\|\eta_i^* - \eta_i\|_2 \leq \varepsilon} \mathrm{KL}(f_{\texttt{clf}}(\mathbf{x} + \eta_i^*),\, \hat{y}), \tag{9}$$

where $\mathrm{KL}(\cdot,\cdot)$ indicates the Kullback-Libler divergence and $\eta_i := \texttt{denoise}(\mathbf{x}+\delta_i)-\mathbf{x}$ is the difference between each denoised image and the original clean image. To find the adversarial perturbation $\eta_i^*$, we perform a $T$-step gradient ascent from each $\eta_i$ with a step size of $2 \cdot \varepsilon/T$, while projecting $\eta_i^*$ to remain within an $\ell_2$-ball of radius $\varepsilon$: *viz.*, the *projected gradient descent* (PGD) (Madry et al., 2018). For the adversarial target $\hat{y}$, we adapt the *consistency* target from the previous robust training method (Jeong et al., 2023) to our denoised smoothing setup by letting the target be the average likelihood of the denoised images, *i.e.*, $\hat{y} := \frac{1}{M} \sum_{i=1}^{M} \texttt{Softmax}\big(f_{\texttt{clf}}(\texttt{denoise}(\mathbf{x} + \delta_i))\big)$.

**Overall training objective.** Building on our proposed training objectives $\mathcal{L}^{\texttt{SCE}}$ and $\mathcal{L}^{\texttt{MAdv}}$, we now present the complete objective for our *Fine-Tuning with Confidence-Aware Denoised Image Selection* (FT-CADIS). Based on our confidence-aware denoised image selection scheme, Confidence-aware selective cross-entropy loss and Confidence-aware masked adversarial loss are applied only to non-hallucinated images $\mathcal{D}_{\mathbf{x},\mathbf{nh}}$ to improve both generalizability and robustness of the smoothed classifier. The overall loss function is as follows:

$$\mathcal{L}^{\texttt{FT-CADIS}} := \mathcal{L}^{\texttt{SCE}} + \lambda \cdot \mathcal{L}^{\texttt{MAdv}}, \tag{10}$$

where $\lambda > 0$ is a hyperparameter, which controls the relative trade-off between the generalizability and the robustness (see Section 4.3). The detailed algorithm for computing our $\mathcal{L}^{\texttt{FT-CADIS}}$ is outlined in Algorithm 1.

**Comparision with CAT-RS.** Our FT-CADIS has drawn motivation from previous confidence-aware training strategies, *e.g.*, CAT-RS (Jeong et al., 2023). The key difference is that FT-CADIS uses the confidence of denoised images based on the pre-trained off-the-shelf classifier while CAT-RS learns their confidence of Gaussian-augmented images during the training of the classifier from scratch. In particular, our method takes advantage of off-the-shelf classifiers which are already capable of providing reasonable confidence for identifying non-hallucinated images. Therefore, we can simply use the non-hallucinated images identified by the off-the-shelf classifiers in our optimization objective. On the other hand, CAT-RS additionally assumes a distribution of semantic-preserving noised sample counts based on the confidence, *i.e.*, average accuracy, of the models currently being trained from scratch. Therefore, the overall confidence remains low especially for complex datasets, resulting in a sub-optimal accuracy of the smoothed classifier (see Table 1b). Our FT-CADIS successfully mitigates this issue based on our carefully designed confidence-based approach utilizing off-the-shelf classifiers, achieving the state-of-the-art robustness even in complex datasets such as ImageNet.

## 4 Experiments

We verify the effectiveness of our proposed training scheme for off-the-shelf classifiers by conducting comprehensive experiments. In Section 4.1, we explain our experimental setups, such as training configurations and evaluation metrics. In Section 4.2, we present the main results on CIFAR-10 and ImageNet. In Section 4.3, we conduct an ablation study to analysis the component-wise effect of our training objective.

### 4.1 Experimental setup

**Baselines.** We mainly consider the following recently proposed methods based on *denoised smoothing* (Salman et al., 2020; Lee, 2021; Carlini et al., 2023; Jeong & Shin, 2024) framework. We additionally compare with other robust training methods for certified robustness based on randomized smoothing (Lecuyer et al., 2019; Cohen et al., 2019; Salman et al., 2019; Jeong & Shin, 2020; Zhai et al., 2020; Horváth et al., 2022a; Yang et al., 2022; Jeong et al., 2021; Horváth et al., 2022b; Jeong et al., 2023). Following the previous works, we consider three different noise levels, $\sigma \in \{0.25, 0.50, 1.00\}$, to obtain smoothed classifiers.

**CIFAR-10 configuration.** We follow the same classifier and the same denoiser employed by Carlini et al. (2023). Specifically, we use the 86M-parameter ViT-B/16 classifier (Dosovitskiy et al., 2020) which is pre-trained and fine-tuned on ImageNet-21K (Deng et al., 2009) and CIFAR-10 (Krizhevsky, 2009), respectively. We use the 50M-parameter 32×32 diffusion model from Nichol & Dhariwal (2021) as the denoiser. We provide more detailed setups in Appendix B.2.

**ImageNet configuration.** We use the 87M-parameter ViT-B/16 classifier which is pre-trained on LAION-2B image-text pairs (Schuhmann et al., 2022) using OpenCLIP (Cherti et al., 2023) and fine-tuned on ImageNet-12K and then ImageNet-1K. Compared to the previous state-of-the-art method, diffusion denoised (Carlini et al., 2023) based on BEiT-large model (Bao et al., 2022) with 305M parameters, we use a much smaller off-the-shelf classifier (30% parameters). We also adopt parameter-efficient fine-tuning with LoRA (Hu et al., 2022), *i.e.*, the number of parameters updated through fine-tuning is only 1% of the total parameters. We use the same denoiser employed by Carlini et al. (2023), *i.e.*, 552M-parameter 256×256 unconditional model from Dhariwal & Nichol (2021). We provide more detailed setups in Appendix B.2.

**Evaluation metrics.** We follow the standard metric in the literature for assessing the certified robustness of smoothed classifiers : the *approximate certified test accuracy* at $r$, which is the fraction of the test set that CERTIFY (Cohen et al., 2019), a practical Monte-Carlo-based certification procedure, classifies correctly with a radius larger than $r$ without abstaining. Throughout our experiments, following Carlini et al. (2023), we use $N = 100,000$ noise samples to certify robustness for entire CIFAR-10 test set and $N = 10,000$ samples for 1,000 randomly selected images from the ImageNet validation set (note that $RS$ methods in Table 1b use $N = 100,000$). We use the hyperparameters from Cohen et al. (2019), specifically $n_0 = 100$ and $\alpha = 0.001$. In ablation study, we additionally consider another standard metric, the *average cerified radius* (ACR) (Zhai

Table 1: CIFAR-10 and ImageNet certified top-1 accuracy. We report the best certified accuracy among the models trained with $\sigma \in \{0.25, 0.50, 1.00\}$, followed by the clean accuracy of the corresponding model in parentheses. RS denotes methods based on randomized smoothing without a denoising procedure, and DS denotes methods based on denoised smoothing. ○ indicates training the classifier with Gaussian-augmented images, ● indicates direct use of the off-the-shelf classifier without fine-tuning, ◑ indicates fine-tuning of the denoiser, ◐ indicates fine-tuning the off-the-shelf classifier, and ◍ indicates parameter-efficient fine-tuning of the off-the-shelf classifier (Hu et al., 2022). The highest certified accuracy in each column is bold-faced. † indicates that extra data is used in the pre-training.

(a) CIFAR-10

| Category | Method | Off-the-shelf | Certified Accuracy at $\varepsilon$ (%) | | | | | |
|---|---|---|---|---|---|---|---|---|
| | | | 0.25 | 0.50 | 0.75 | 1.00 | 1.25 | 1.50 |
| RS | PixelDP (Lecuyer et al., 2019) | ○ | (71.0)22.0 | (44.0)2.0 | - | - | - | - |
| | Gaussian (Cohen et al., 2019) | ○ | (77.0)61.0 | (66.0)43.0 | (66.0)32.0 | (66.0)22.0 | (47.0)17.0 | (47.0)14.0 |
| | SmoothAdv (Salman et al., 2019) | ○ | (85.0)73.0 | (76.0)58.0 | (75.0)48.0 | (57.0)38.0 | (53.0)33.0 | (53.0)29.0 |
| | Consistency (Jeong & Shin, 2020) | ○ | (77.8)68.8 | (75.8)58.1 | (72.9)48.5 | (52.3)37.8 | (52.3)33.9 | (52.3)29.9 |
| | MACER (Zhai et al., 2020) | ○ | (81.0)71.0 | (81.0)59.0 | (66.0)46.0 | (66.0)38.0 | (66.0)29.0 | (45.0)25.0 |
| | Boosting (Horváth et al., 2022a) | ○ | (83.4)70.6 | (76.8)60.4 | (71.6)52.4 | (52.4)38.8 | (52.4)34.4 | (52.4)**30.4** |
| | DRT (Yang et al., 2022) | ○ | (81.5)70.4 | (72.6)60.2 | (71.9)50.5 | (56.1)39.8 | (56.4)**36.0** | (56.4)**30.4** |
| | SmoothMix (Jeong et al., 2021) | ○ | (77.1)67.9 | (77.1)57.9 | (74.2)47.7 | (61.8)37.2 | (61.8)31.7 | (61.8)25.7 |
| | ACES (Horváth et al., 2022b) | ○ | (77.6)69.0 | (73.4)57.2 | (73.4)47.0 | (57.0)37.8 | (57.0)32.2 | (57.0)27.8 |
| | CAT-RS (Jeong et al., 2023) | ○ | (76.3)68.1 | (76.3)58.8 | (76.3)48.2 | (62.3)38.5 | (62.3)32.7 | (62.3)27.1 |
| DS | Denoised (Salman et al., 2020) | ◑ | (72.0)56.0 | (62.0)41.0 | (62.0)28.0 | (44.0)19.0 | (42.0)16.0 | (44.0)13.0 |
| | Score-based Denoised (Lee, 2021) | ● | 60.0 | 42.0 | 28.0 | 19.0 | 11.0 | 6.0 |
| | Diffusion Denoised† (Carlini et al., 2023) | ● | (88.1)76.7 | (88.1)63.0 | (88.1)45.3 | (77.0)32.1 | - | - |
| | Diffusion Denoised†1 (Carlini et al., 2023) | ◐ | (91.2)79.3 | (91.2)65.5 | (91.2)48.7 | (81.5)35.5 | - | - |
| | Multi-scale Denoised† (Jeong & Shin, 2024) | ◑ | - | (90.3)61.9 | - | (85.1)32.9 | - | (79.6)16.2 |
| | **FT-CADIS (Ours)†** | ◐ | (88.7)**80.3** | (88.7)**68.4** | (88.7)**54.5** | (74.9)**39.9** | (74.9)31.6 | (74.9)23.5 |

(b) ImageNet

| Category | Method | Off-the-shelf | Certified Accuracy at $\varepsilon$ (%) | | | | |
|---|---|---|---|---|---|---|---|
| | | | 0.50 | 1.00 | 1.50 | 2.00 | 2.50 |
| RS | PixelDP (Lecuyer et al., 2019) | ○ | (33.0)16.0 | - | - | - | - |
| | Gaussian (Cohen et al., 2019) | ○ | (67.0)49.0 | (57.0)37.0 | (57.0)29.0 | (44.0)19.0 | (44.0)15.0 |
| | SmoothAdv (Salman et al., 2019) | ○ | (65.0)56.0 | (55.0)45.0 | (55.0)38.0 | (42.0)28.0 | (42.0)26.0 |
| | Consistency (Jeong & Shin, 2020) | ○ | (55.0)50.0 | (55.0)44.0 | (55.0)34.0 | (41.0)24.0 | (41.0)21.0 |
| | MACER (Zhai et al., 2020) | ○ | (68.0)57.0 | (64.0)43.0 | (64.0)31.0 | (48.0)25.0 | (48.0)18.0 |
| | Boosting (Horváth et al., 2022a) | ○ | (68.0)57.0 | (57.0)44.6 | (57.0)38.4 | (44.6)28.6 | (38.6)24.6 |
| | DRT (Yang et al., 2022) | ○ | (52.2)46.8 | (49.8)44.4 | (49.8)39.8 | (49.8)30.4 | (49.8)29.0 |
| | SmoothMix (Jeong et al., 2021) | ○ | (55.0)50.0 | (55.0)43.0 | (55.0)38.0 | (40.0)26.0 | (40.0)24.0 |
| | ACES (Horváth et al., 2022b) | ○ | (63.2)54.0 | (55.4)42.2 | (55.0)35.6 | (39.2)25.6 | (50.6)22.0 |
| | CAT-RS (Jeong et al., 2023) | ○ | (44.0)38.0 | (44.0)35.0 | (44.0)31.0 | (44.0)27.0 | (44.0)24.0 |
| DS | Denoised (Salman et al., 2020) | ◐ | (60.0)33.0 | (38.0)14.0 | (38.0)6.0 | - | - |
| | Score-based Denoised (Lee, 2021) | ● | 41.0 | 24.0 | 11.0 | - | - |
| | Diffusion Denoised† (Carlini et al., 2023) | ● | (82.8)71.1 | (77.1)54.3 | (77.1)38.1 | (60.0)29.5 | - |
| | Multi-scale Denoised† (Jeong & Shin, 2024) | ◑ | (76.6)54.6 | (76.6)39.8 | (76.6)23.0 | (69.0)14.6 | - |
| | **FT-CADIS (Ours)†** | ◍ | (81.1)**71.9** | (77.0)**60.1** | (77.0)**45.8** | (66.2)**39.4** | (66.2)**30.7** |

et al., 2020): the average of cerified radii on the test set $D_{test}$ while assigning incorrect samples as 0: *viz.*, $\text{ACR} := \frac{1}{|D_{test}|} \sum_{(\mathbf{x}, y) \in D_{test}} [\text{CR}(f, \sigma, \mathbf{x}) \cdot \mathbb{1}_{g(\mathbf{x})=y}]$, where $\text{CR}(\cdot)$ denotes the lower bound of certified radius CERTIFY returns.

## 4.2 Main experiments

**Results on CIFAR-10.** In Table 1a, we compare the performance of the baselines and FT-CADIS on CIFAR-10. Overall, FT-CADIS outperforms all existing state-of-the-art denoised smoothing (denoted by DS) approaches in every radii. For example, our method improves the best-performing denoised smoothing

---
[1]Further fine-tune the classifier on denoised images from CIFAR-10.

Table 2: Comparison of the architectures and parameters between the previous state-of-the-art certified defense methods and FT-CADIS on ImageNet.

| Method | CAT-RS (Jeong et al., 2023) | Diffusion Denoised (Carlini et al., 2023) | Multi-scale Denoised (Jeong & Shin, 2024) | **FT-CADIS (Ours)** |
|---|---|---|---|---|
| Denoiser | - | Guided Diffusion (Dhariwal & Nichol, 2021) | Guided Diffusion (Dhariwal & Nichol, 2021) | Guided Diffusion (Dhariwal & Nichol, 2021) |
| Classifier | ResNet-50 (He et al., 2016) | BEiT-large (Bao et al., 2022) | ViT-B/16 (Dosovitskiy et al., 2020) | ViT-B/16 (+LoRA) (Dosovitskiy et al., 2020) |
| Parameters | Denoiser : - 
 Classifier : 26M | Denoiser : 552M 
 Classifier : 305M | Denoiser : 552M 
 Classifier : 87M | Denoiser : 552M 
 Classifier : 87M |
| Trainable | Denoiser : - 
 Classifier : 26M | Denoiser : - 
 Classifier : - | Denoiser : 552M 
 Classifier : - | Denoiser : - 
 Classifier : **0.9M** |

Table 3: Comparison of ACR and certified accuracy for ablations of $\mathcal{L}^{\text{FT-CADIS}}$ on CIFAR-10 with $\sigma = 1.00$.

| Fine-tuning objective design | ACR | Certified Accuracy at $\varepsilon$ (%) | | | | | | |
|---|---|---|---|---|---|---|---|---|
| | | 0.00 | 0.25 | 0.50 | 0.75 | 1.00 | 1.25 | 1.50 |
| $\mathcal{L}^{\text{SCE}} + \lambda \cdot \mathcal{L}^{\text{MAdv}}$ ($\mathcal{L}^{\text{FT-CADIS}}$; **Ours**) | **0.784** | 48.1 | 43.5 | **40.6** | **36.9** | **32.5** | **28.6** | **23.7** |
| (w/o) Non-hallucinated condition of $\mathcal{L}^{\text{SCE}}$ | 0.726 | 52.4 | 45.6 | 40.4 | 35.9 | 31.2 | 26.1 | 21.9 |
| (w/o) Mask of $\mathcal{L}^{\text{MAdv}}$ | 0.374 | 11.2 | 10.9 | 10.4 | 10.2 | 10.2 | 10.2 | 10.2 |
| Cross-entropy loss $\mathcal{L}^{\text{CE}}$ (Carlini et al., 2023) | 0.633 | **54.4** | **45.8** | 39.3 | 33.2 | 28.1 | 22.4 | 17.3 |

method (Carlini et al., 2023) by $35.5\% \rightarrow 39.9\%$ at $\varepsilon = 1.00$. FT-CADIS also outperforms every randomzied smoothing techinque up to a radius of $\varepsilon \leq 1.00$. Even though our method slightly underperforms at higher radii in terms of certified accuracy, we note that FT-CADIS is the only denoised smoothing method which achieves a reasonable robustness at $\varepsilon > 1.00$. This means that our FT-CADIS effectively alleviates the distribution shift and hallucination issues observed in previous methods based on denoised smoothing (Carlini et al., 2023). We provide the detailed results in Appendix B.5.

**Results on ImageNet.** In Table 1b, we compare the performance of the baselines and FT-CADIS on ImageNet, which is a far more complex dataset than CIFAR-10. In summary, FT-CADIS outperforms all existing state-of-the-art methods in every radii. In particular, our method surpasses the certified accuracy of diffusion denoised (Carlini et al., 2023) by $9.9\%$ at $\varepsilon = 2.00$. In Table 2, we also compare the architecture and trainable parameters of each method. Our method even shows remarkable parameter efficiency, *i.e.*, we only update 0.9M parameters, which is 3% of Jeong et al. (2023) and 0.2% of Jeong & Shin (2024). The overall results highlight the scalability of FT-CADIS, indicating its effectiveness in practical applications with only a small parameter updates. We provide the detailed results in Appendix B.5 and further discuss the efficiency of LoRA (Hu et al., 2022) on FT-CADIS in Appendix F.

## 4.3 Ablation study

In this section, we conduct an ablation study to further analyze the design of our proposed losses, the impact of updating the set of non-hallucinated images, and the component-wise effectiveness of our method. Unless otherwise stated, we report the test results based on a randomly sampled 1,000 images from the CIFAR-10 test set.

**Effect of overall loss design.** Table 3 presents a comparison of variants of $\mathcal{L}^{\text{FT-CADIS}}$, including: (a) removing the non-hallucinated condition of $\mathcal{L}^{\text{SCE}}$ in Eq. (8), (b) removing the masking condition of $\mathcal{L}^{\text{MAdv}}$ in Eq. (9), and (c) training with cross-entropy loss $\mathcal{L}^{\text{CE}}$ only. In summary, we observe that (a) using only non-hallucinated images for $\mathcal{L}^{\text{SCE}}$ achieves better ACR and effectively balances between accuracy and robustness. Additionally, we find that (b) the mask "$\mathcal{D}_{\mathbf{x},\mathbf{nh}} = M$" in $\mathcal{L}^{\text{MAdv}}$ is crucial for stable training, as it prevents the optimization of adversarial images that have lost the semantic of the original image; and (c) FT-CADIS

Table 4: Comparison of ACR and certified accuracy for the ablation of the update of $\mathcal{D}_{\mathbf{x},\mathbf{nh}}$ on CIFAR-10.

| Noise | Update of $\mathcal{D}_{\mathbf{x},\mathbf{nh}}$ | ACR | Certified Accuracy at $\varepsilon$ (%) | | | | | | |
|---|---|---|---|---|---|---|---|---|---|
| | | | 0.00 | 0.25 | 0.50 | 0.75 | 1.00 | 1.25 | 1.50 |
| $\sigma = 0.25$ | ✗ | 0.632 | **91.1** | **80.4** | 66.7 | 49.0 | 0.0 | 0.0 | 0.0 |
| | ✓ | **0.642** | 87.9 | 78.7 | **68.0** | **54.0** | 0.0 | 0.0 | 0.0 |
| $\sigma = 0.50$ | ✗ | 0.765 | **75.4** | **66.9** | 56.0 | 46.0 | 36.2 | 28.7 | 21.6 |
| | ✓ | **0.806** | 72.2 | 64.1 | **57.2** | **48.1** | **40.3** | **34.1** | **25.9** |
| $\sigma = 1.00$ | ✗ | 0.626 | **53.4** | **45.9** | 38.2 | 32.9 | 27.3 | 22.5 | 16.4 |
| | ✓ | **0.783** | 48.1 | 43.5 | **40.6** | **36.9** | **32.4** | **28.5** | **23.8** |

Table 5: Comparison of ACR and certified accuracy for ablations of $\mathcal{L}^{\mathtt{MAdv}}$ on CIFAR-10 with $\sigma = 0.50$. Every design adopts the $\mathbb{1}[|\mathcal{D}_{\mathbf{x},\mathbf{nh}}| = M]$ masking condition.

| Adversarial objective design | ACR | Certified Accuracy at $\varepsilon$ (%) | | | | | | |
|---|---|---|---|---|---|---|---|---|
| | | 0.00 | 0.25 | 0.50 | 0.75 | 1.00 | 1.25 | 1.50 |
| (a) $\max_{i,\eta_i^*} \mathrm{KL}(f_{\mathtt{clf}}(\mathbf{x} + \eta_i^*), y)$ | 0.802 | 71.7 | 64.3 | 56.2 | 48.0 | 39.8 | 33.8 | 25.7 |
| (b) $\frac{1}{M} \sum_i (\max_{\eta_i^*} \mathrm{KL}(f_{\mathtt{clf}}(\mathbf{x} + \eta_i^*), \hat{y}))$ | 0.792 | **74.9** | **65.8** | 56.1 | 47.8 | 39.7 | 31.8 | 23.4 |
| (c) $\frac{1}{M} \sum_i (\max_{\eta_i^*} \mathrm{KL}(f_{\mathtt{clf}}(\mathbf{x} + \eta_i^*), y))$ | 0.792 | 74.8 | 64.9 | 57.0 | 48.0 | 39.9 | 31.5 | 23.0 |
| $\max_{i,\eta_i^*} \mathrm{KL}(f_{\mathtt{clf}}(\mathbf{x} + \eta_i^*), \hat{y})\,(\mathcal{L}^{\mathtt{MAdv}}; \textbf{Ours})$ | **0.806** | 72.2 | 64.1 | **57.2** | **48.1** | **40.3** | **34.1** | **25.9** |

demonstrates higher robustness and ACR by combining Confidence-aware selective cross-entropy loss and Confidence-aware masked adversarial loss.

**Effect of Confidence-aware masked adversarial loss design.** We further investigate the components of Confidence-aware masked adversarial loss. Table 5 presents three variants of $\mathcal{L}^{\mathtt{MAdv}}$ in Eq. (9): (a) replacing the consistent target $\hat{y}$ with the assigned label $y$, (b) substituting the outer maximization with an average-case, and (c) combining both (a) and (b). Overall, we find that our proposed $\mathcal{L}^{\mathtt{MAdv}}$ demonstrates superior ACR compared to the variants, achieving the highest certified robustness while maintaining satisfactory clean accuracy. It shows that both design choices, *i.e.*, maximizing loss over adversarial images and using soft-labeled adversarial targets, are particularly effective.

**Effect of iterative update of $\mathcal{D}_{\mathbf{x},\mathbf{nh}}$.** Our FT-CADIS iteratively updates the set of non-hallucinated images, *i.e.*, $\mathtt{denoise}(\mathbf{x} + \delta) \in \mathcal{D}_{\mathbf{x},\mathbf{nh}}$, to deal with the distribution shift from the pre-training distribution (clean images) to fine-tuning distribution (denoised images). Table 4 shows the effect of $\mathcal{D}_{\mathbf{x},\mathbf{nh}}$ on varying $\sigma \in \{0.25, 0.50, 1.00\}$. For all noise levels, the iterative update strategy shows higher ACR with higher robustness. We find that the fine-tuning classifier increases the ratio of applying $\mathcal{L}^{\mathtt{MAdv}}$ (see Figure 5 in Appendix D), *i.e.*, $f_{\mathtt{clf}}$ gradually classifies all the denoised images of $\mathbf{x}$ correctly, thereby focusing on maximizing robustness and achieving a better trade-off between accuracy and robustness (Zhang et al., 2019a).

**Effect of $\lambda$.** In the fine-tuning objective of FT-CADIS in Eq. (10), $\lambda$ determines the ratio between $\mathcal{L}^{\mathtt{MAdv}}$ and $\mathcal{L}^{\mathtt{SCE}}$. Figure 3a illustrates how $\lambda$ affects the certified accuracy across different radii, with $\lambda$ varying in $\{0.5, 1.0, 2.0, 4.0, 8.0\}$ and $\sigma = 0.50$. As $\lambda$ increases, the robustness at high radii improves although the clean accuracy decreases, *i.e.*, the trade-off between clean accuracy and robustness.

**Effect of $M$.** Figure 3b shows the impact of $M$ on the model when varying $M \in \{1, 2, 4, 8\}$. The robustness of the smoothed classifier improves as $M$ increases, while the clean accuracy decreases. With a higher $M$, the model is exposed to more denoised images included in $\mathcal{D}_{\mathbf{x},\mathbf{nh}}$, reducing the distribution shift from clean images to denoised images. This increases the confidence of the smoothed classifier, *i.e.*, the accuracy on denoised images, resulting in more robust predictions.

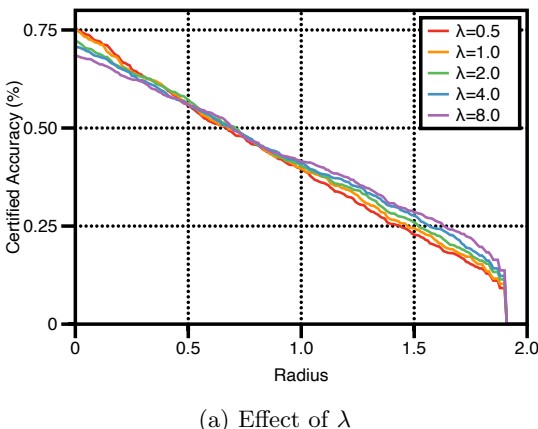
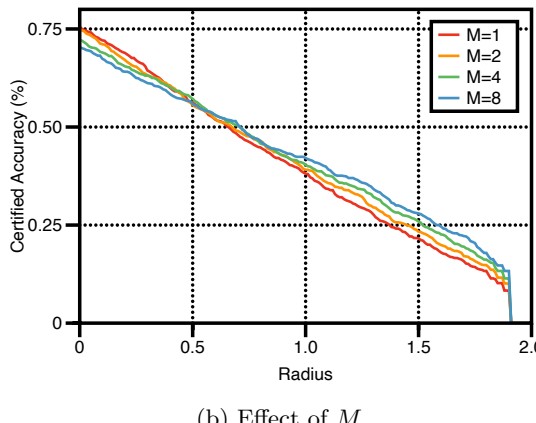

(a) Effect of $\lambda$                                                  (b) Effect of $M$

Figure 3: Comparison of certified accuracy for components in FT-CADIS, (a) $\lambda$ and (b) $M$, on CIFAR-10. We plot the results at $\sigma = 0.50$. We provide detailed results in Appendix C.

## 5 Related Work

**Certified adversarial robustness.** Recently, various defenses have been proposed to build robust classifiers against adversarial attacks. In particular, *certified defenses* have gained significant attention due to their guarantee of robustness (Wong & Kolter, 2018; Wang et al., 2018a;b; Wong et al., 2018). Among them, *randomized smoothing* (Lecuyer et al., 2019; Li et al., 2019; Cohen et al., 2019) shows the state-of-the-art performance by achieving the tight certified robustness guarantee over $\ell_2$-adversary (Cohen et al., 2019). This approach converts any base classifier, *e.g.*, a neural network, into a provably robust smoothed classifier by taking a majority vote over random Gaussian noise. To maximize the robustness of the smoothed classifier, the base classifier should be trained with Gaussian-augmented images (Lecuyer et al., 2019; Cohen et al., 2019; Salman et al., 2019; Zhai et al., 2020; Jeong & Shin, 2020; Jeong et al., 2023). For instance, Salman et al. (2019) employed adversarial training (Madry et al., 2018) within the randomized smoothing framework, while Jeong & Shin (2020) suggested training a classifier using simple consistency regularization. Moreover, Jeong et al. (2023) introduced sample-wise control of target robustness, motivated by the accuracy-robustness trade-off (Tsipras et al., 2019; Zhang et al., 2019a) in smoothed classifiers. However, training base classifiers specifically for Gaussian-augmented images requires large training costs and thus these methods suffer from scalability issues in complex datasets, *e.g.*, the accuracy drops severely in the ImageNet dataset.

**Denoised smoothing.** Denoised smoothing alleviates the aforementioned scalability issue of randomized smoothing by introducing "denoise-and-classify" strategy. This approach allows randomized smoothing to be applied to any off-the-shelf classifier trained on clean images, *i.e.*, not specifically trained on Gaussian-augmented images, by adding a denoising step before feeding Gaussian-augmented images into the classifier. In recent years, diffusion probabilistic models have emerged as an ideal choice for the denoiser in the denoised smoothing scheme. In particular, Lee (2021) have initially explored the applicability of diffusion models in denoised smoothing, and Carlini et al. (2023) further observe that combining the latest diffusion models with an off-the-shelf classifier provides a state-of-the-art design for certified robustness. Meanwhile, Jeong & Shin (2024) investigate the trade-off between robustness and accuracy in denoised smoothing, and proposed a multi-scale smoothing scheme that incorporates denoiser fine-tuning.

Our work aims to improve the certified robustness of smoothed classifiers in denoised smoothing, which is determined by the average accuracy of the off-the-shelf classifiers under denoised images. We improve such robustness by addressing hallucination and distribution shift issues of denoised images. Specifically, we focus on filtering out hallucinated images based on the confidence of off-the-shelf classifiers, and then fine-tune off-the-shelf classifiers with non-hallucinated images.

# 6 Conclusion

We propose FT-CADIS, a scalable fine-tuning strategy of off-the-shelf classifiers for certified robustness. Specifically, we propose to use the *confidence* of off-the-shelf classifiers to mitigate the intrinsic drawbacks of the denoised smoothing framework, *i.e.*, hallucination and distribution shift. We also demonstrate that this can be achieved by updating only 1% of the total parameters. We hope that our method could be a meaningful step for the future research to develop a scalable approach for certified robustness.

**Limitation and future work.** In this work, we apply an efficient training technique for off-the-shelf classifiers based on LoRA (Hu et al., 2022). Nevertheless, certification remains a bottleneck for throughput, due to its majority voting process involving a large number of forward inferences, *i.e.*, $N = 100,000$. An important future work would be to accelerate the certification process for a more practical deployment of our method. In addition, certain public vision APIs do not allow us to access the underlying off-the-shelf classifiers, *i.e.*, black-box. In such cases, our method is not directly applicable, and further research on training methods that are independent of model parameters, such as prompt-tuning (Jia et al., 2022), will be necessary.

# Acknowledgements

This work was conducted by Center for Applied Research in Artificial Intelligence (CARAI) grant funded by Defense Acquisition Program Administration (DAPA) and Agency for Defense Development (ADD) (UD230017TD), and supported by Institute for Information & communications Technology Promotion (IITP) grant funded by the Korea government (MSIT) (No.RS-2019-II190075, Artificial Intelligence Graduate School Program (KAIST); No. RS-2019-II190079, Artificial Intelligence Graduate School Program (Korea University)), and by Culture, Sports and Tourism R&D Program through the Korea Creative Content Agency grant funded by the Ministry of Culture, Sports and Tourism in 2024 (Project Name: International Collaborative Research and Global Talent Development for the Development of Copyright Management and Protection Technologies for Generative AI, Project Number: RS-2024-00345025).

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

# Supplementary Material

**Appendix: Confidence-aware Denoised Fine-tuning of
Off-the-shelf Models for Certified Robustness**

## A  Training Procedure of FT-CADIS

---
**Algorithm 1** Fine-Tuning with Confidence-Aware Denoised Image Selection (FT-CADIS)

---
**Require:** training sample $(\mathbf{x}, y)$. variance of Gaussian noise $\sigma$. number of noises $M$. off-the-shelf classifier $f_{\texttt{clf}}$. attack $\ell_2$-norm $\varepsilon > 0$. adversarial target $\hat{y} \in \Delta^{K-1}$. coefficient of Confidence-aware masked adversarial loss $\lambda > 0$.

---
1: Generate $\hat{\mathbf{x}}_1 = \textsc{NoiseAndDenoise}(\mathbf{x}_1, \sigma), \cdots, \hat{\mathbf{x}}_M = \textsc{NoiseAndDenoise}(\mathbf{x}_M, \sigma)$   ▷ $\mathbf{x}_i$: copy of $\mathbf{x}$
2: Identify $\mathcal{D}_{\mathbf{x},\mathbf{nh}} = \{\hat{\mathbf{x}}_i \mid f_{\texttt{clf}}(\hat{\mathbf{x}}_i) = y, i \in [1, ..., M]\}$
3: **for** $i = 1$ to $M$ **do**
4:     $\mathcal{L}_i \leftarrow \mathbb{CE}(f_{\texttt{clf}}(\hat{\mathbf{x}}_i), y)$
5:     $\eta_i^* \leftarrow \underset{\|\eta_i^* - \eta_i\|_2 \leq \varepsilon}{\arg\max} \; \mathrm{KL}(f_{\texttt{clf}}(\mathbf{x} + \eta_i^*), \hat{y}), \; \eta_i := \hat{\mathbf{x}}_i - \mathbf{x}$
6: **end for**
7: $\mathcal{L}_{1:M}^{\pi}, \textsc{indices} \leftarrow \texttt{argsort}(\mathcal{L}_{1:M}), \; \mathcal{D}_{\mathbf{x},\mathbf{nh}}^{\pi} \leftarrow \{\hat{\mathbf{x}}_{\textsc{indices.index}(i)}^{\pi} \mid \hat{\mathbf{x}}_i \in \mathcal{D}_{\mathbf{x},\mathbf{nh}}\}$
8: **if** $\mathcal{D}_{\mathbf{x},\mathbf{nh}}^{\pi} \neq \emptyset$ **then**
9:     $\mathcal{L}^{\texttt{SCE}} \leftarrow \frac{1}{M}(\sum_{\hat{\mathbf{x}}_i^{\pi} \in \mathcal{D}_{\mathbf{x},\mathbf{nh}}^{\pi}} \mathcal{L}_i^{\pi})$
10: **else**
11:     $\mathcal{L}^{\texttt{SCE}} \leftarrow \frac{1}{M}(\mathcal{L}_1^{\pi})$   ▷ $\mathcal{L}_1^{\pi}$: lowest cross-entropy loss
12: **end if**
13: $\mathcal{L}^{\texttt{MAdv}} \leftarrow \mathbb{1}[|\mathcal{D}_{\mathbf{x},\mathbf{nh}}| = M] \cdot \underset{i}{\max} \, \mathrm{KL}(f_{\texttt{clf}}(\mathbf{x} + \eta_i^*), \, \hat{y})$
14: $\mathcal{L}^{\texttt{FT-CADIS}} \leftarrow \mathcal{L}^{\texttt{SCE}} + \lambda \cdot \mathcal{L}^{\texttt{MAdv}}$

---

---
**Algorithm 2** Noise-and-Denoise Procedure (Carlini et al., 2023)

---
1: **function** $\textsc{NoiseAndDenoise}(\mathbf{x}, \sigma)$:
2:     $t^*, \alpha_{t^*} \leftarrow \textsc{GetTimestep}(\sigma)$
3:     $\mathbf{x}_{t^*} \leftarrow \sqrt{\alpha_{t^*}}(\mathbf{x} + \delta), \; \delta \sim \mathcal{N}(0, \sigma^2 \boldsymbol{I})$
4:     $\hat{\mathbf{x}} \leftarrow \texttt{denoise}(\mathbf{x}_{t^*}; t^*)$   ▷ $\texttt{denoise}$ : one-shot diffusion denoising process
5:     **return** $\hat{\mathbf{x}}$
6: **end function**
7:
8: **function** $\textsc{GetTimestep}(\sigma)$:
9:     $t^* \leftarrow$ find the timestep $t$ s.t. $\sigma^2 = \frac{1 - \alpha_t}{\alpha_t}$   ▷ $\alpha_t$ : noise level constant of diffusion model
10:     **return** $t^*, \alpha_{t^*}$
11: **end function**

---

# B Experimental Details

## B.1 Datasets

**CIFAR-10** (Krizhevsky, 2009) consists of 60,000 RGB images of size 32×32, with 50,000 images for training and 10,000 for testing. Each image is labeled as one of 10 classes. We apply the standard data augmentation, including random horizontal flip and random translation up to 4 pixels, as used in previous works (Cohen et al., 2019; Salman et al., 2019; Zhai et al., 2020; Jeong & Shin, 2020; Jeong et al., 2021; 2023). No additional normalization is applied except for scaling the pixel values from [0,255] to [0.0, 1.0] when converting image into a tensor. The full dataset can be downloaded at https://www.cs.toronto.edu/ kriz/cifar.html.

**ImageNet** (Russakovsky et al., 2015) consists of 1.28 million training images and 50,000 validation images, each labeled into one of 1,000 classes. For the training images, we apply 224×224 randomly resized cropping and horizontal flipping. For the test images, we resize them to 256×256 resolution, followed by center cropping to 224×224. Similar to CIFAR-10, no additional normalization is applied except for scaling the pixel values to [0.0, 1.0]. The full dataset can be downloaded at https://image-net.org/download.

## B.2 Training

**Noise-and-Denoise Procedure.** We follow the protocol of Carlini et al. (2023) to obtain the denoised images for fine-tuning. Firstly, the given image $\mathbf{x}$ is clipped to the range [-1,1] as expected by the off-the-shelf diffusion models. Then, the perturbed image is obtained from a certain diffusion time step according to the target noise level. Finally, we adopt a one-shot denoising, *i.e.*, outputting the best estimate for the denoised image in a single step, resulting in a denoised image within the range of [-1,1]. Since this range differs from the typical range of [0, 1] assumed in prior works, we set the target noise level to twice the usual level for training and certification. A detailed implementation can be found at https://github.com/ethz-spylab/diffusion-denoised-smoothing and the algorithm is provided in Algorithm 2.

**CIFAR-10 fine-tuning.** We conduct an end-to-end fine-tuning of a pre-trained ViT-B/16 (Dosovitskiy et al., 2020), considering different scenarios of $\sigma \in \{0.25, 0.50, 1.00\}$ for randomized smoothing. The same $\sigma$ is applied to both the training and certification. As part of the data pre-processing, we interpolate the dataset to 224×224. Our fine-tuning follows the common practice of supervised ViT training. The default setting is shown in Table 6a. We use the linear *lr* scaling rule (Goyal et al., 2017): *lr = base lr × batch size ÷ 256*. The batch size is calculated as *batch per GPU × number of GPUs × accum iter // number of noises*, where *accum iter* denotes the batch accumulation hyperparameter.

**ImageNet fine-tuning.** We adopt LoRA (Hu et al., 2022) to fine-tune a pre-trained ViT-B/16 (Dosovitskiy et al., 2020) in a parameter-efficient manner. We use the same training scenarios as for CIFAR-10. As part of the data pre-processing, we interpolate the dataset to 384×384. The default setting is shown in Table 6b. Compared to end-to-end fine-tuning, we reduce the regularization setup, *e.g.*, weight decay, *lr* decay, drop path, and gradient clipping. For LoRA fine-tuning, we freeze the original model except for the classification layer. Then, LoRA weights are incorporated into each query and value projection matrix of the self-attention layers of ViT. For these low-rank matrices, we use Kaiming-uniform initialization for weight $\boldsymbol{A}$ and zeros for weight $\boldsymbol{B}$, following the official code. To implement LoRA with ViT, we refer to https://github.com/JamesQFreeman/LoRA-ViT.

## B.3 Hyperparameters

In our proposed loss functions (see Eqs. (8), (9), and (10)), there are two main hyperparameters: the coefficient $\lambda$ for the Confidence-aware masked adversarial loss, and the attack radius $\varepsilon$ of Confidence-aware masked adversarial loss. We have determined the optimal configurations for two hyperparameters through a simple grid search on $\lambda$ over [1,2,4] and $\varepsilon$ over [0.125, 0.25, 0.5, 1.0].

For CIFAR-10, we use $\lambda = 1.0, 2.0, 4.0$ for $\sigma = 0.25, 0.50, 1.00$, respectively. Assuming that $\texttt{denoise}(\mathbf{x} + \delta) \approx \mathbf{x}$ with high probability, we adopt a small $\varepsilon = 0.25$ by default, which is increased to 0.50 after 10 epochs only for $\sigma = 1.00$. For ImageNet, we use $\lambda = 2.0, 1.0, 2.0$ for $\sigma = 0.25, 0.50, 1.00$ respectively, and $\varepsilon$ is fixed

Table 6: Denoised fine-tuning settings for the off-the-shelf classifier on CIFAR-10 and ImageNet.

(a) CIFAR-10 end-to-end fine-tuning

| Configuration | Value |
|---|---|
| Optimizer | AdamW (Loshchilov & Hutter, 2019) |
| Optimizer momentum | $\beta_1, \beta_2 = 0.9, 0.999$ |
| Base learning rate | 5e-4 ($\sigma = 0.25, 0.50$), 1e-4 ($\sigma = 1.00$) |
| Weight decay | start, end = 0.04, 0.4 (cosine schedule) |
| Layer-wise lr decay (Clark et al., 2020; Bao et al., 2022) | 0.65 |
| Batch size | 128 |
| Learning rate schedule | cosine decay (Loshchilov & Hutter, 2022) |
| Warmup epochs (Goyal et al., 2017) | 3 |
| Training epochs | 30 (early stopping at 20) |
| Drop path (Huang et al., 2016) | 0.2 |
| Gradient clipping (Zhang et al., 2019b) | 0.3 |

(b) ImageNet LoRA (Hu et al., 2022) fine-tuning

| Configuration | Value |
|---|---|
| Optimizer | AdamW (Loshchilov & Hutter, 2019) |
| Optimizer momentum | $\beta_1, \beta_2 = 0.9, 0.999$ |
| Base learning rate | 2e-4 ($\sigma = 0.25$), 4e-4 ($\sigma = 0.50, 1.00$) |
| Weight decay | start, end = 0.02, 0.2 ($\sigma = 0.25$) start, end = 0.01, 0.1 ($\sigma = 0.50, 1.00$) |
| Layer-wise lr decay (Clark et al., 2020; Bao et al., 2022) | 0.8 ($\sigma = 0.25$), 0.9 ($\sigma = 0.50, 1.00$) |
| Batch size | 128 |
| Learning rate schedule | cosine decay (Loshchilov & Hutter, 2022) |
| Warmup epochs (Goyal et al., 2017) | 1 |
| Training epochs | 10 (early stopping at 5) |
| Drop path (Huang et al., 2016) | 0.0 |
| Gradient clipping (Zhang et al., 2019b) | 1.0 |
| LoRA rank r | 4 |
| LoRA scaler $\alpha$ | 4 |

at 0.25 for all noise levels. Although the number of noises $M$ and the number of attack steps $T$ can also be tuned for better performance, we fix $M = 4$ and $T = 4$ for CIFAR-10. For ImageNet, we fix $M = 2$ and $T = 1$ to reduce the overall training cost. Additional training configurations are provided in Table 6. Due to the extensive training cost of large models, we have adjusted some training configurations for the ablation study, $e.g.$, the warmup and training epochs are reduced to 2 and 20, with $\varepsilon$ doubled after 10 epochs.

## B.4 Computing infrastructure

In summary, we conduct our experiments using NVIDIA GeForce RTX 2080 Ti GPUs for CIFAR-10, NVIDIA GeForce RTX 3090 and NVIDIA RTX A6000 GPUs for ImageNet. In the CIFAR-10 experiments, we utilize 4 NVIDIA GeForce RTX 2080 Ti GPUs for fine-tuning per run, resulting in ~8 hours of training cost. During the certification, we use 7 NVIDIA GeForce RTX 2080 Ti GPUs for data splitting, taking ~9 minutes per image (with $N = 100,000$ for each inference) to perform a single pass of smoothed inference. In the ImageNet experiments, we utilize 4 NVIDIA RTX A6000 GPUs for fine-tuning per run, observing ~51 hours of training cost. During the certification, 8 NVIDIA GeForce RTX 3090 GPUs are used in parallel, taking ~4 minutes per image (with $N = 10,000$ for each inference) to complete a single pass of smoothed inference.

### B.5 Detailed results on main experiments

Table 7: Certified accuracy of FT-CADIS for varying levels of Gaussian noise $\sigma$ on CIFAR-10 and ImageNet. Values in bold-faced indicate the ones reported in Table 1a for CIFAR-10 and Table 1b for ImageNet.

(a) CIFAR-10

| Noise | Certified Accuracy at $\varepsilon$ (%) | | | | | | |
|---|---|---|---|---|---|---|---|
| | 0.00 | 0.25 | 0.50 | 0.75 | 1.00 | 1.25 | 1.50 |
| $\sigma = 0.25$ | 88.7 | **80.3** | **68.4** | **54.5** | 0.0 | 0.0 | 0.0 |
| $\sigma = 0.50$ | 74.9 | 67.3 | 58.7 | 49.2 | **39.9** | **31.6** | **23.5** |
| $\sigma = 1.00$ | 49.6 | 45.5 | 41.0 | 36.8 | 32.5 | 28.4 | 24.2 |

(b) ImageNet

| Noise | Certified Accuracy at $\varepsilon$ (%) | | | | | |
|---|---|---|---|---|---|---|
| | 0.00 | 0.50 | 1.00 | 1.50 | 2.00 | 2.50 |
| $\sigma = 0.25$ | 81.1 | **71.9** | 0.0 | 0.0 | 0.0 | 0.0 |
| $\sigma = 0.50$ | 77.0 | 69.3 | **60.1** | **45.8** | 0.0 | 0.0 |
| $\sigma = 1.00$ | 66.2 | 60.7 | 54.0 | 46.4 | **39.4** | **30.7** |

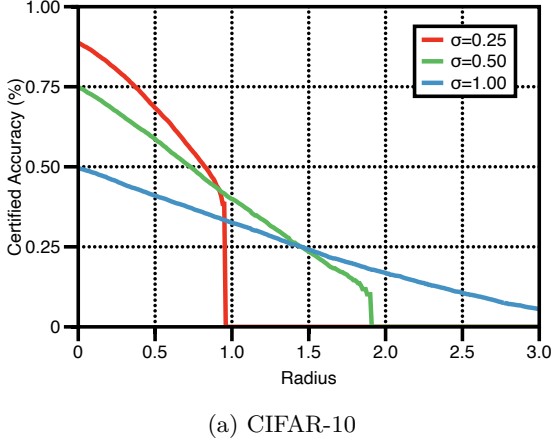
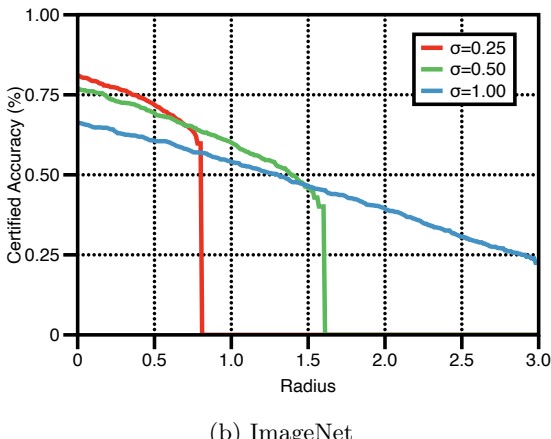

(a) CIFAR-10         (b) ImageNet

Figure 4: Certified accuracy of FT-CADIS at different levels of Gaussian noise $\sigma \in \{0.25, 0.50, 1.00\}$. Upper bounds in radius are calculated with $N = 100{,}000$ for CIFAR-10 and $N = 10{,}000$ for ImageNet.

## C Detailed Results on Effect of $\lambda$ and $M$

Table 8: Comparison of ACR and certified accuracy for ablations of varying $\lambda$ on CIFAR-10 with $\sigma = 0.50$.

| Setups | ACR | Certified Accuracy at $\varepsilon$ (%) | | | | | | |
|---|---|---|---|---|---|---|---|---|
| | | 0.00 | 0.25 | 0.50 | 0.75 | 1.00 | 1.25 | 1.50 |
| $\lambda = 0.50$ | 0.786 | **75.7** | **64.9** | 55.8 | 47.5 | 39.5 | 30.8 | 22.8 |
| $\lambda = 1.00$ | 0.797 | 75.3 | 64.3 | 56.3 | 47.9 | 39.7 | 32.8 | 24.5 |
| $\lambda = 2.00$ | 0.806 | 72.2 | 64.1 | **57.2** | **48.1** | 40.3 | 34.1 | 25.9 |
| $\lambda = 4.00$ | 0.814 | 70.9 | 63.3 | 55.9 | 48.0 | 41.0 | 35.0 | 27.7 |
| $\lambda = 8.00$ | **0.823** | 68.6 | 62.4 | 56.0 | 47.6 | **41.5** | **35.9** | **28.5** |

Table 9: Comparison of ACR and certified accuracy for ablations of varying $M$ on CIFAR-10 with $\sigma = 0.50$.

| Setups | ACR | Certified Accuracy at $\varepsilon$ (%) | | | | | | |
|---|---|---|---|---|---|---|---|---|
| | | 0.00 | 0.25 | 0.50 | 0.75 | 1.00 | 1.25 | 1.50 |
| $M = 1$ | 0.773 | **75.6** | **67.0** | 56.3 | 46.2 | 38.1 | 29.4 | 21.6 |
| $M = 2$ | 0.790 | 74.9 | 65.4 | 55.6 | 47.6 | 39.1 | 32.2 | 23.3 |
| $M = 4$ | 0.806 | 72.2 | 64.1 | **57.2** | **48.1** | 40.3 | 34.1 | 25.9 |
| $M = 8$ | **0.817** | 70.4 | 62.5 | 55.9 | 47.9 | **42.1** | 35.8 | 27.9 |

## D Effect of Iterative Update of $\mathcal{D}_{\mathbf{x,nh}}$ on $\mathcal{L}^{\mathtt{MAdv}}$

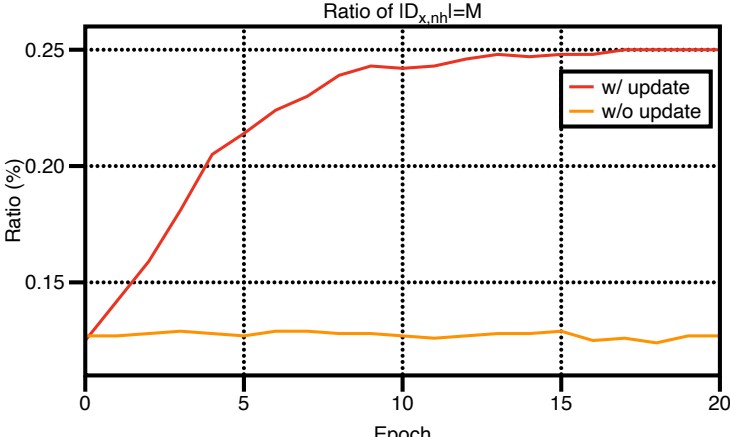

Figure 5: Change in the ratio of $|\mathcal{D}_{\mathbf{x,nh}}| = M$, *i.e.*, ratio of clean images $\mathbf{x}$ satisfying the masking condition of $\mathcal{L}^{\mathtt{MAdv}}$, during fine-tuning on CIFAR-10 with $\sigma = 1.00$, depending on whether $\mathcal{D}_{\mathbf{x,nh}}$ is being updated or not. In the legend, red indicates that $\mathcal{D}_{\mathbf{x,nh}}$ is iteratively updated, while orange indicates that $\mathcal{D}_{\mathbf{x,nh}}$ is fixed.

# E Details on Certifying Robustness of FT-CADIS

We simply follow the common evaluation framework of the baselines (Carlini et al., 2023; Jeong & Shin, 2024). In denoised smoothing framework (Salman et al., 2020), the robustness of the smoothed classifier $g$ is guaranteed based on the accuracy of the off-the-shelf classifier $f_{\texttt{clf}}$ under denoised images, $viz.$, $\mathbb{P}_\delta[f_{\texttt{clf}}(\texttt{denoise}(\mathbf{x} + \delta)) = y] =: p_g(\mathbf{x}, y)$. However, since $f_{\texttt{clf}}$ is a high-dimensional neural network, this accuracy cannot be computed directly. Instead, we estimate it using the practical Monte-Carlo based algorithm CERTIFY from Cohen et al. (2019).

CERTIFY algorithm consists of two main procedures: (a) given input $\mathbf{x}$, identifying the most probable output class $\hat{c}_A$ of $f$, and (b) computing a lower-bound on the probability that output of $f$ is $\hat{c}_A$:

(a) Using a small number $n_0$ ($e.g.$, $n_0 = 100$) of denoised images and taking a majority vote over the outputs of $f_{\texttt{clf}}$, $i.e.$, computing $f_{\texttt{clf}}(\texttt{denoise}(\mathbf{x} + \delta))$ $n_0$ times to identify the most frequent class $\hat{c}_A$.

(b) Using a large number $n$ ($e.g.$, $n = 100{,}000$) of denoised images to estimate the lower bound of $\mathbb{P}_\delta[f_{\texttt{clf}}(\texttt{denoise}(\mathbf{x} + \delta)) = \hat{c}_A]$, $viz.$, $\underline{p_g}(\mathbf{x}, \hat{c}_A)$, considering the significance level $\alpha$ (see Cohen et al. (2019) for details).

Finally, the certified $\ell_2$-norm radius of smoothed classifier, derived by Cohen et al. (2019), is calculated as $\sigma \cdot \Phi^{-1}(\underline{p_g}(\mathbf{x}, \hat{c}_A))$ when $\underline{p_g}(\mathbf{x}, \hat{c}_A)) > \frac{1}{2}$; otherwise return ABSTAIN. It verifies that the smoothed classifier $g$ does not change its output within an $\ell_2$ ball of radius $\sigma \cdot \Phi^{-1}(\underline{p_g}(\mathbf{x}, \hat{c}_A))$ around any input $\mathbf{x}$.

## F   Analysis on Efficiency of FT-CADIS with LoRA

Table 10: Comparison of fine-tuning costs and ACR between the baselines (Carlini et al., 2023; Jeong & Shin, 2024) and FT-CADIS on ImageNet with $\sigma = 0.50$. † indicates that the numbers are taken from the original paper.

| Method | LoRA (Hu et al., 2022) | ACR | Fine-tuned model | Trainable parameters | GPU days |
|---|---|---|---|---|---|
| Diffusion Denoised (Carlini et al., 2023) | - | 0.896 | - | 0M | - |
| Multi-scale Denoised† (Jeong & Shin, 2024) | ✗ | 0.743 | Guided Diffusion (Dhariwal & Nichol, 2021) | 552M | 32 |
| **FT-CADIS (Ours)** | ✗ | 1.013 | ViT-B/16 (Dosovitskiy et al., 2020) | 87M | 11.2 |
| | ✓ | 1.001 | ViT-B/16 (Dosovitskiy et al., 2020) | 0.9M | 8.4 |

In Table 10, we compare the time complexity of different methods. Firstly, our FT-CADIS (without LoRA (Hu et al., 2022)) largely outperforms Multi-scale Denoised (Jeong & Shin, 2024), *i.e.*, $0.743 \rightarrow 1.013$ in ACR, with significantly smaller training costs, *i.e.*, $32 \rightarrow 11.2$ in GPU days. Furthermore, LoRA reduces the training time of FT-CADIS by 25% and the trainable parameters by 99% compared to full parameter fine-tuning, while maintaining ACR on par with full parameter fine-tuning. We note that the efficiency of LoRA can be further improved through advancements of GPU infrastructure or low-level code optimization. Since LoRA is generally applicable to other robust training techniques, we hope that our work initiates the research direction on alleviating the large cost of robust training.

Meanwhile, Diffusion Denoised (Carlini et al., 2023) proposes to obtain a robust classifier *without* fine-tuning. While they achieve reasonable robustness in an extremely efficient manner, *i.e.*, no fine-tuning costs, they suffer from the fundamental limitation associated with hallucination effect of the denoiser (see Figure 2a). Due to this bottleneck, we find that the robustness of Diffusion Denoised degrades particularly at large radii (see Table 1b). One of our main contributions is identifying and addressing such hallucination issue, achieving improved robustness, *i.e.*, $0.896 \rightarrow 1.013$ in ACR.

## G  Analysis on Pre-trained Classifiers

Table 11: Comparison of ACR and certified accuracy between FT-CADIS and Diffusion Denoised (Carlini et al., 2023) on CIFAR-10 with $\sigma = 0.50$.

| Method | Classifier | Test Accuracy | ACR | Certified Accuracy at $\varepsilon$ (%) | | | | | | |
|---|---|---|---|---|---|---|---|---|---|---|
| | | | | 0.00 | 0.25 | 0.50 | 0.75 | 1.00 | 1.25 | 1.50 |
| Diffusion Denoised (Carlini et al., 2023) | ResNet-110 (1.7M) | 93.7% | 0.669 | **75.0** | 62.8 | 50.2 | 38.9 | 30.9 | 22.8 | 15.6 |
| **FT-CADIS (Ours)** | ResNet-110 (1.7M) | 93.7% | 0.754 | 68.8 | 60.9 | 53.9 | 45.2 | 38.3 | 29.4 | 23.4 |
| | ViT-B/16 (85.8M) | **97.9%** | **0.806** | 72.2 | **64.1** | **57.2** | **48.1** | **40.3** | **34.1** | **25.9** |

In Table 11, we investigate the relationship between our proposed method and pre-trained classifiers. The results show that our proposed method still outperforms Carlini et al. (2023) on ResNet-110 (He et al., 2016), *i.e.*, a much smaller architecture than ViT-B/16 (Dosovitskiy et al., 2020). Also, we observe that the certified robustness of our method improves as we use more advanced pre-trained classifiers, *e.g.*, FT-CADIS based on ViT-B/16 largely improves the results based on ResNet-110. These findings demonstrate that the effectiveness of our method is not restricted to specific classifiers and can be further enhanced with continuous advancements in this field.

## H  Additional Results on $\ell_\infty$ Adversarial Attack

Table 12: Comparison of $\ell_\infty$ certified accuracy (%) on CIFAR-10 with radius $\varepsilon$. We report the model with the highest certified $\ell_\infty$ accuracy for each method.

| CIFAR-10 ($\ell_\infty$) | Diffusion Denoised (Carlini et al., 2023) | Multi-scale Denoised (Jeong & Shin, 2024) | **FT-CADIS (Ours)** |
|---|---|---|---|
| Robust ($\varepsilon = \frac{2}{255}$) | 62.9 | 67.1 | **71.8** |

In Table 12, we present the certified robustness of defense methods on another threat model, *i.e.*, $\ell_\infty$-norm. Specifically, we leverage the geometric relationships between the $\ell_2$-norm ball and $\ell_\infty$-norm ball to assess the robustness under $\ell_\infty$-norm (Salman et al., 2019). Through this simple conversion, our proposed method can provide robustness against other $\ell_p$-adversaries.

## I  Additional Examples of Hallucinated Images

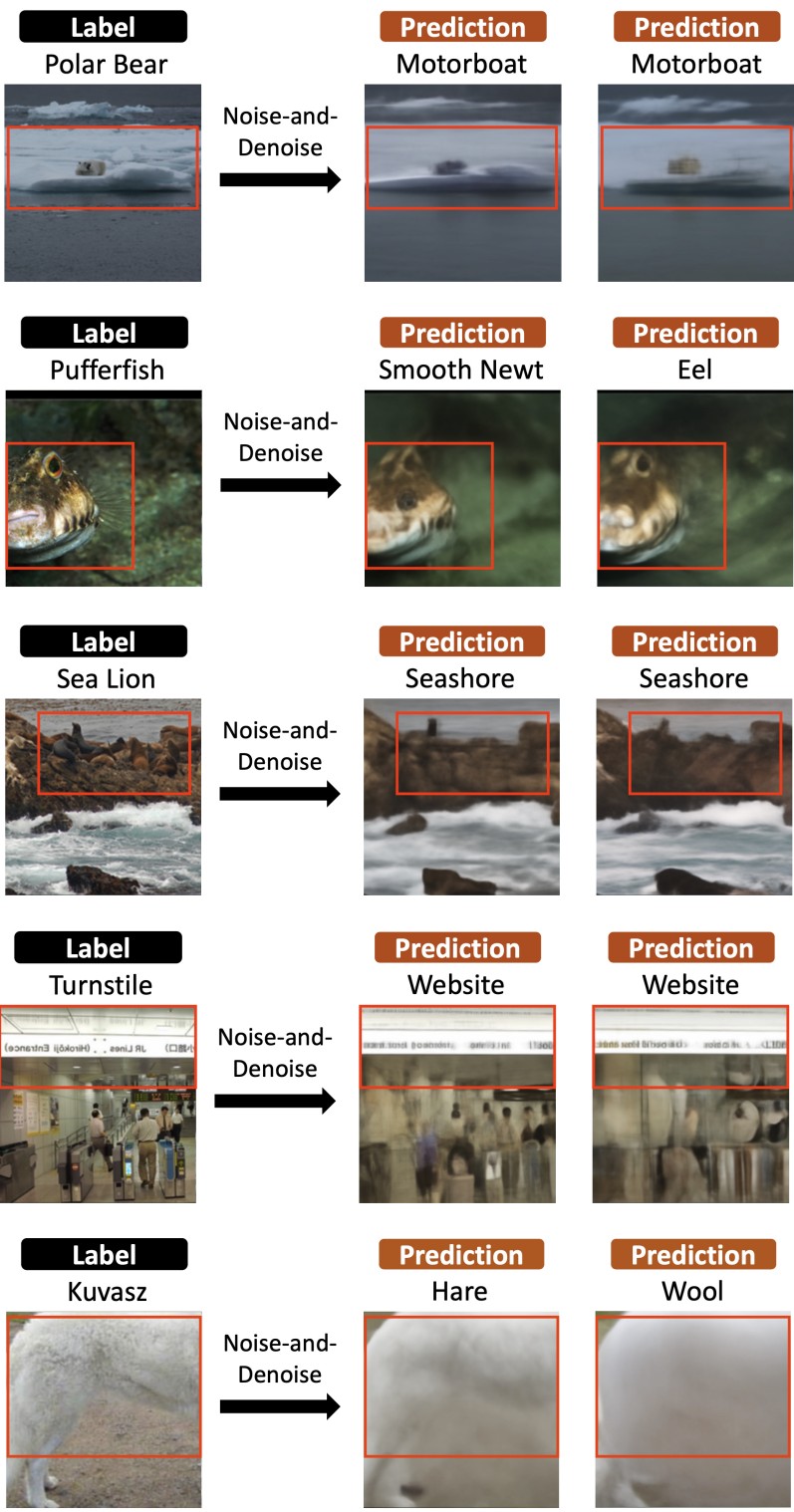

Figure 6: Additional examples of hallucinated images after the noise-and-denoise procedure on ImageNet at $\sigma = 1.00$. The red box indicates the areas where the original semantic of the image is corrupted.

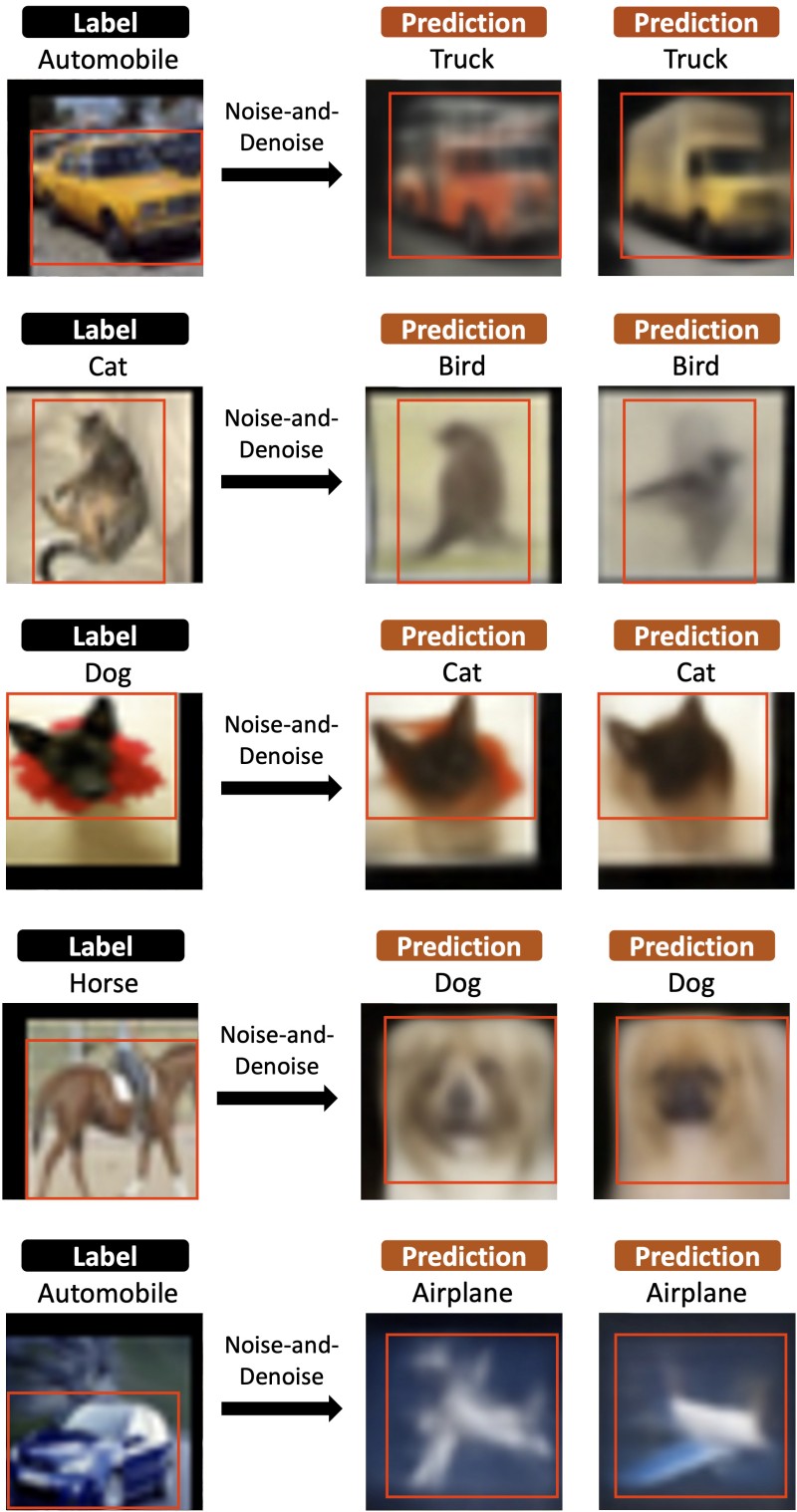

Figure 7: Additional examples of hallucinated images after the noise-and-denoise procedure on CIFAR-10 at $\sigma = 1.00$. The red box indicates the areas where the original semantic of the image is corrupted.

## J  Analysis on Training Stability of FT-CADIS

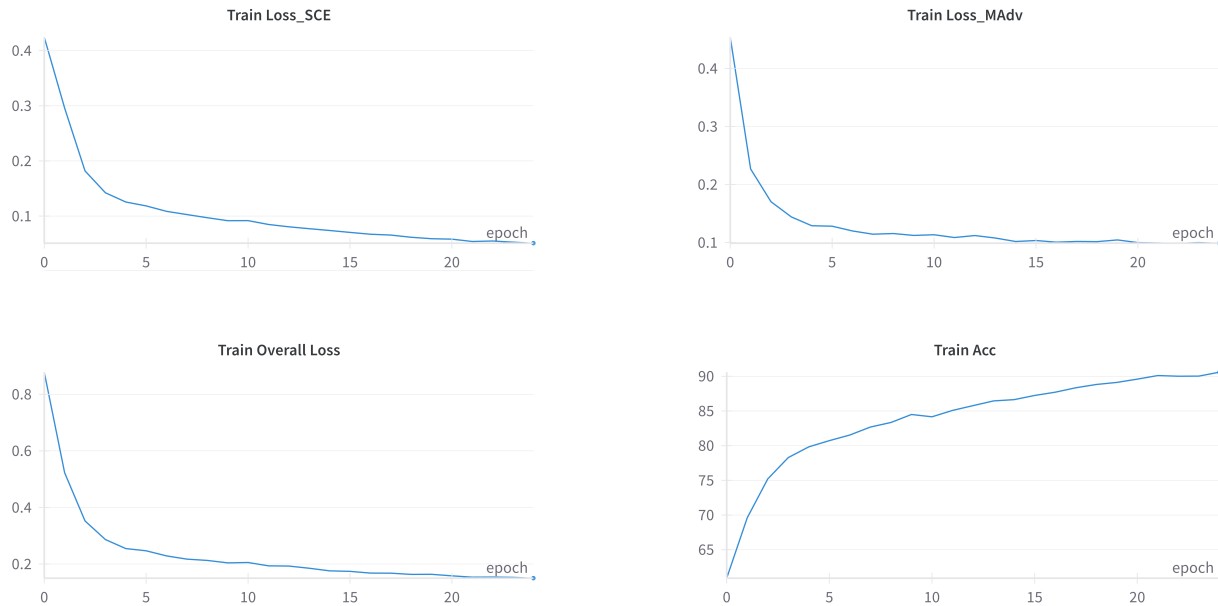

Figure 8: Plots of (1) Confidence-aware selective cross-entropy loss $\mathcal{L}^{\texttt{SCE}}$, (2) Confidence-aware masked adversarial loss $\mathcal{L}^{\texttt{MAdv}}$, (3) Overall training objective $\mathcal{L}^{\texttt{FT-CADIS}}$, and (4) Top-1 accuracy from our main experiments on CIFAR-10 with $\sigma = 0.25$.

In this section, we demonstrate that our training objective $\mathcal{L}^{\texttt{FT-CADIS}}$ remains stable throughout the fine-tuning process. As mentioned in Section 3.3, our overall objective is composed of Confidence-aware selective cross-entropy loss $\mathcal{L}^{\texttt{SCE}}$ and Confidence-aware masked adversarial loss $\mathcal{L}^{\texttt{MAdv}}$. In Figure 8, we show that (1) the training loss, including the adversarial loss, converges smoothly without oscillation, and (2) the training accuracy also converges well.

