# OpenReview forum: "Confidence-aware Denoised Fine-tuning of Off-the-shelf Models for Certified Robustness"
_TMLR — Accepted by TMLR_

### Review · Reviewer_u4XJ · 2024-08-29

**Summary Of Contributions:**

This work introduces a method called FT-CADIS (Fine-Tuning with Confidence-Aware Denoised Image Selection), which aims to enhance the certified robustness of off-the-shelf classifiers against adversarial attacks. The key contributions include the introduction of a confidence-aware selective cross-entropy loss and a confidence-aware masked adversarial loss, specifically designed to handle hallucinated images generated by denoising procedures. This approach focuses on improving both the generalizability and robustness of classifiers by selecting non-hallucinated images for fine-tuning. The method achieves improvements in certified robustness with minimal computational overhead by updating 1% of the classifier's parameters, using techniques like LoRA for parameter-efficient fine-tuning. Experiments on benchmarks like CIFAR-10 and ImageNet demonstrate that FT-CADIS outperforms existing denoised smoothing methods across various adversary radii, especially in high Gaussian variance regimes.

**Audience:**

Yes

**Broader Impact Concerns:**

There is no broader impact concern.

**Claims And Evidence:**

Yes

**Requested Changes:**

To address the identified weaknesses, the following modifications are suggested to strengthen the paper:

1. (**Suggestion**): It would be better to include a rigorous theoretical foundation, particularly by providing certified robustness guarantees if possible. This is a key expectation in adversarial defense research and would significantly enhance the credibility and impact of the proposed method. A theoretical analysis or proof that aligns the FT-CADIS approach with certified robustness frameworks should be developed and presented in the paper.

2. While the method shows strong performance against $\ell_2$-norm adversaries, it is critical to evaluate and demonstrate its effectiveness against other adversarial threat models, such as $\ell_\infty$-norm, $\ell_1$-norm, trace-norm, or even more complex attack models. Including these evaluations would provide a more comprehensive understanding of the method's generalizability and robustness across different types of adversarial attacks.

3. The paper should address the dependency on pre-trained classifiers by either reducing this reliance or by providing a detailed analysis of how the method performs across different pre-trained models, including those that may not be robust or well-calibrated. This analysis would help in understanding the limitations and potential risks associated with this dependency and propose strategies to mitigate them.

4. The paper should simplify the implementation details where possible and provide clear guidelines or best practices for implementing the confidence-aware selection and masked adversarial loss steps. This would reduce the barrier to entry for other researchers and practitioners who wish to adopt the method. Additionally, the paper should include more detailed information on the tuning process required for these steps to ensure that the method can be effectively implemented and replicated.

**Strengths And Weaknesses:**

Strength:

- This paper presents an innovative methodology where the confidence-aware image selection strategy marks a significant improvement over traditional denoised smoothing techniques, effectively mitigating the issue of hallucination and distribution shift.
- The method achieves high certified robustness across multiple benchmarks, demonstrating its effectiveness in improving model resilience to adversarial attacks.
- Additionally, by only updating 1% of the classifier's parameters, the method is computationally efficient, which is crucial for deploying robust models in resource-constrained environments.
- The comprehensive experimental validation provided in the submission, including comparisons with a wide range of baselines, thorough ablation studies, and analysis of the method's scalability, is another strong aspect.

However, the method has some weaknesses.

- While the approach is intuitive, it lacks a rigorous theoretical foundation, such as providing certified robustness guarantees that are commonly expected in adversarial defense research.
- This work also does not delve deeply into the theoretical analysis of the method's effectiveness, which could have strengthened the overall contribution.
- While it performs well against $\ell_2$-norm adversaries, it is unclear how it generalizes to other types of adversarial attacks, such as $\ell_\infty$-norm, $\ell_1$-norm, trace-norm or more complex threat models.
- The approach also relies heavily on the availability and quality of pre-trained classifiers, which could compromise its effectiveness if these classifiers are not robust or well-calibrated.
- Furthermore, the method involves several intricate steps, such as confidence-aware selection and masked adversarial loss, which might be challenging to implement and require careful tuning.

---

> ### Author Response · Authors · 2024-09-28
> **Response to Reviewer u4XJ**
>
> Dear reviewer u4XJ,
>
> We sincerely appreciate your efforts in reviewing our manuscript. We respond to each comment in the following content. In the revised manuscript, we have marked the revisions with “red”.
>
> ---
> **[W1/C1] Certified robustness guarantees are not provided.**
>
> We clarify that our method does provide certified robustness guarantees. Specifically, our fine-tuning method works upon the randomized smoothing [1] framework, which already has an established protocol to obtain certified robustness against $\ell_2$-adversary [1] (see Eq. (3) in our manuscript). For your information, we added more details about the certification of robustness in Appendix D of our revised manuscript.
>
> [1] Cohen et al., Certified adversarial robustness via randomized smoothing, ICML 2019
>
> ---
> **[W2] The theoretical analysis of the effectiveness of FT-CADIS is insufficient.**
>
> Our method is designed to improve the theoretical certified robustness of (denoised) smoothed classifiers, which is calculated by the average accuracy of the classifiers under denoised images. To improve such theoretical guarantees, our FT-CADIS addresses issues of hallucination and distribution shift, which have caused the accuracy drop in denoised images. We included this discussion in Section 4.4 of our revised manuscript.
>
> ---
> **[W3/C2] Demonstrate the generalizability of FT-CADIS to other adversarial threat models.**
>
> We focus on randomized smoothing [1] (which gives $\ell_2$-robustness) as it is known as the state-of-the-art approach on certifying adversarial robustness. However, smoothed classifiers obtained from our method can also certify against other adversaries,  e.g., $\ell_\infty$-norm (see the table below). We included the results and the corresponding discussion in Appendix G of our revised manuscript.
>
> \begin{array}{l|cc|c}
> \hline
> \text{CIFAR-10 ($\ell_\infty$)} & \text{Diffusion Denoised [2]} & \text{Multi-scale Denoised [3]} & \textbf{FT-CADIS (Ours)} \newline
> \hline
> \text{Robust ($\varepsilon=\frac{2}{255}$)} & 62.9 & 67.1 & \textbf{71.8} \newline
> \hline
> \end{array}
>
> [1] Cohen et al., Certified adversarial robustness via randomized smoothing, ICML 2019 \
> [2] Carlini et al., (Certified!!) Adversarial Robustness for Free!, ICLR 2023 \
> [3] Jeong et al., Multi-scale Diffusion Denoised Smoothing, NeurIPS 2023
>
> ---
> **[W4/C3] Strong dependency on the availability and quality of pre-trained classifiers.**
>
> Thank you for the opportunity to clarify this point. The classifiers do not need to be robust by default, as denoised smoothing provides the robustness to the (non-robust) classifiers.  In the table below, our proposed method still outperforms the main baseline [1] on ResNet-110,  i.e., a much smaller architecture than ViT-B/16, demonstrating that the effectiveness of our method is not limited to specific pre-trained classifiers. Also, we observe that the certified robustness of our method improves as we use more advanced pre-trained classifiers, e.g., FT-CADIS based on ViT-B/16 largely improves the results based on ResNet-110. We added the results and corresponding discussion in Appendix F of our revised manuscript.
>
> \begin{array}{lccc|cccccccc}
> \hline
> \phantom{Method} & \phantom{Classifier} & \phantom {Test Accuracy} & \phantom{ACR} &  \rlap{~~~~~~~~~~\text{Certified Accuracy at $\varepsilon$ (\\%)}} \newline
> \hline
> \text{Method} & \text{Classifier} & \text{Test Accuracy} & \text{ACR} & \text{0.00} & \text{0.25} & \text{0.50} & \text{0.75} & \text{1.00} & \text{1.25} & \text{1.50} \newline
> \hline
> \text{Diffusion Denoised [1]} & \text{ResNet-110 (1.7M)} & \text{93.7\\%} & \text{0.669} & \textbf{75.0} & \text{62.8} & \text{50.2} & \text{38.9} & \text{30.9} & \text{22.8} & \text{15.6} \newline
> \hline
> \textbf{FT-CADIS (Ours)} & \text{ResNet-110 (1.7M)} & \text{93.7\\%} & \text{0.754} & \text{68.8} & \text{60.9} & \text{53.9} & \text{45.2} & \text{38.3} & \text{29.4} & \text{23.4} \newline
> \textbf{FT-CADIS (Ours)} & \text{ViT-B/16 (85.8M)} & \textbf{97.9\\%} & \textbf{0.806} & \text{72.2} & \textbf{64.1} & \textbf{57.2} & \textbf{48.1} & \textbf{40.3} & \textbf{34.1} & \textbf{25.9} \newline
> \hline
> \end{array}
>
> [1] Carlini et al., (Certified!!) Adversarial Robustness for Free!, ICLR 2023
>
> ---
> **[W5/C4] Implementation details need to be simplified, and more information on the tuning process should be provided.**
>
> In our experiments, we did not put much efforts in tuning hyperparameters: we have tuned only two main hyperparameters (a) $\lambda$ in [1, 2, 4] and (b) $\varepsilon$ in [0.125, 0.25, 0.5, 1.0] and other hyperparameters are fixed (see Table 6).
>
> Regarding the implementation, we have provided the details of confidence-aware selection (see Section 3.2) and masked adversarial loss steps (see Section 3.3). To provide further details, we uploaded our official code at [this link](https://anonymous.4open.science/r/ft-cadis-CE32) and added more details of the tuning process in Appendix B.3 of our revised manuscript.

---

> > ### Comment · Reviewer_u4XJ · 2024-09-30
> >
> > Thanks to the authors for the response, revision, and shared code. I checked the revision and code. My concerns have been well addressed and I have no further concerns.

---

> > > ### Author Response · Authors · 2024-09-30
> > > **Response to Reviewer u4XJ**
> > >
> > > Dear reviewer u4XJ,
> > >
> > > Thank you for your comments and for taking the time to review our manuscript. We are pleased to hear that our replies have resolved your concerns.
> > >
> > > If you have any further comments or suggestions, please let us know. We are committed to improving the quality of our work, and we value your feedback.
> > >
> > > Thank you very much,\
> > > Authors

---

### Review · Reviewer_wc3e · 2024-09-16

**Summary Of Contributions:**

The paper proposes a novel fine-tuning scheme, FT-CADIS, for image classifiers to improve the certified robustness obtained by employing them in the denoised smoothing framework. FT-CADIS first fine-tunes the classifier only on images which are still correctly classified after the noise-and-denoise procedure. Moreover, it integrates adversarial training when the classifier performs well on the denoised images. This approach aims at improving both clean performance and robustness, which yield higher average certified radius. In the experiments on CIFAR-10 and ImageNet, FT-CADIS mostly outperforms the baselines.

**Audience:**

Yes

**Broader Impact Concerns:**

No concerns.

**Claims And Evidence:**

Yes

**Requested Changes:**

I think the discussion about efficiency should be updated, as well as the presentation improved (see details above).

**Strengths And Weaknesses:**

Strengths
- The proposed method provides a simple way to deal with the hallucination coming from the denoising model. Moreover, it integrates adversarial training to further improve the obtained guarantees.

- The experimental results show the effectiveness of FT-CADIS, which mostly outperforms the baselines. Several ablation studies are shown to justify the design choices.

Weaknesses
- The claims about better efficiency of the proposed fine-tuning are a bit weak: first, LoRA is a standard method and could be applied to the other baselines as well, and it is not clear how (if) it'd change their performance. Second, there are baselines like Carlini et al. (2023) which don't need any fine-tuning and can be seen as more efficient than the proposed method. Finally, the cost of adversarial training and discarding the hallucinated images is not taken into account. While it's not straightforward to compare the exact computational cost of each method, the current discussion seems at least incomplete, potentially misleading.

- The presentation can be largely improved:
  - The method by which the adversarial points $\eta_i^*$ are computed is not discussed until the appendix. This is a major part of the proposed algorithm which I think should be included in the main part.
  - Similarly, in Sec. 3.2 it's mentioned that the classifier $f_\texttt{clf}$ is iteratively updated to renew the training set, but it's not clear how. Is it simply the fine-tuned model at the current iteration or something else?
  - The notations $f_\texttt{clf}$ and $F_\texttt{clf}$ seem to be used interchangeably. If they indicate the same model, a single notation should be used, if not, the difference should be defined.
  - Table 4, Table 5, Fig. 3 and Fig. 4 appear in the opposite order compared to the text, which is very inconvenient for the reader.
  - Sec. 2: "most provable output" --> "most probable output"?
  - Sec. 4.3 discusses the difference to the training scheme of the main experiments, which have however not been introduced before.

- In the adversarial loss (Eq. (9)), I see the role of having the mask (the indicator function), but can it make training unstable? The adversarial loss, which might by high, is in fact added when all the denoised images are correctly classified, i.e. clean loss is low. Moreover, the adversarial loss might be applied or not for consecutive iterations, which it seems might make the overall loss quite largely oscillate.

Overall, I think the proposed method is reasonable and supported by the experimental results, but the paper would benefit from more complete discussion about efficiency and a clearer presentation.

---

> ### Author Response · Authors · 2024-09-28
> **Response to Reviewer wc3e**
>
> Dear reviewer wc3e,
>
> We sincerely appreciate your efforts in reviewing our manuscript. We respond to each comment in the following content. In the revised manuscript, we have marked the revisions with “red”.
>
> ---
> **[W1]  The claims about efficiency of the proposed fine-tuning with LoRA are weak.**
>
> We sincerely appreciate your valuable comments and the opportunity to clarify this point.  First, we would like to emphasize that previous methods are conceptually different from our approach, and thus fine-tuning classifiers with LoRA [1] is not applicable to them. Second, while [2] does not require fine-tuning, it does not show enough robustness, particularly at large $\ell_2$ radii, due to the hallucination and distribution shift issues (see Table 1(b)). Lastly, in the table below, we compare the computational costs of our method with the main baseline [3] on ImageNet. Our method indeed shows speed-ups of 2.9x (full parameter update) and 3.8x (LoRA) compared to [3]. We included these detailed discussions in Appendix E of our revised manuscript.
>
> \begin{array}{lcccc}
> \hline
> \text{Method} & \text{ACR} & \text{Model} & \text{Trainable parameters} &  \text{GPU days} \newline
> \hline
> \text{Multi-scale Denoised [3] } & \text{0.743} & \text{Guided Diffusion [4]} & \text{552M}  & \text{32} \newline
> \hline
> \text{FT-CADIS (Full parameter)} & \text{1.013} & \text{ViT-B/16 [5]} & \text{87M} &  \text{11.2} \newline
> \hline
> \text{FT-CADIS (LoRA [1] )} & \text{1.001} & \text{ViT-B/16 [5]} & \text{0.9M} &  \text{8.4} \newline
> \hline
> \end{array}
>
> [1] Hu et al., LoRA: Low-Rank Adaptation of Large Language Models, ICLR 2022 \
> [2] Carlini et al., (Certified!!) Adversarial Robustness for Free!, ICLR 2023 \
> [3] Jeong et al., Multi-scale Diffusion Denoised Smoothing, NeurIPS 2023 \
> [4] Dhariwal & Nichol et al., Diffusion models beat GANs on image synthesis, NeurIPS 2021 \
> [5] Dosovitskiy et al., An Image is Worth 16x16 Words: Transformers for Image Recognition at Scale, ICLR 2021
>
> ---
> **[W2] Editorial comments.**
>
> We sincerely appreciate your constructive feedback and the thorough review of our manuscript. We have carefully considered each of your points and incorporated the following changes into our revised manuscript:
>
> - How are the adversarial points $\eta^*_i$ computed? : Added clarification in Section 3.3.
> - How the classifier $f_{clf}$ is iteratively updated? : Clarified in Section 3.2.
> - Notations $f_{clf}$ and $F_{clf}$ are confused : Unified to $f_{clf}$ throughout the manuscript.
> - Tables and Figures are in the opposite order compared to the text : Modified the layout accordingly.
> - Sec.2: “most provable output” → “most probable output” :  Corrected typo in Section 2.
> - Sec.4.3: Setups of main experiments are not introduced before : Clarified in Appendix B.3
>
> ---
> **[W3] Can masked adversarial loss make training unstable?**
>
> Thank you for the opportunity to clarify this point. Based on extensive experiments, we found that training with our overall objective, which includes adversarial loss, remains stable. We also note that our method shows stable results, i.e., certified accuracy, under various choices of hyperparameters (see Tables 8 and 9).  Nevertheless, for your information, we have included the training loss graph from one of our main experiments (i.e., CIFAR-10 with $\\sigma = 0.25$), which shows that the training loss converges smoothly without oscillation, and corresponding analysis in Appendix I of our revised manuscript.

---

> ### Comment · Reviewer_wc3e · 2024-10-10
>
> I thank the authors for the response and additional experiments. I think the clarity of the paper has improved.
>
> > fine-tuning classifiers with LoRA [1] is not applicable to them
>
> Could the author clarify why this is the case?
>
> > Second, while [2] does not require fine-tuning, it does not show enough robustness, particularly at large radii, due to the hallucination and distribution shift issues (see Table 1(b))
>
> [2] is actually the strongest baseline in most cases in Table 1, and it even outperforms FT-CADIS at small radii when using the same classifier, as shown in the new Table 11. Then, I think it shouldn't be dismissed as *"it does not show enough robustness"*, but rather it shows a trade-off between computational cost and performance (it's more efficient than the proposed FT-CADIS with, on average, slightly worse robustness).

---

> > ### Author Response · Authors · 2024-10-13
> > **Response to Reviewer wc3e**
> >
> > Dear reviewer wc3e,
> >
> > We sincerely appreciate your additional comments. We respond to each comment in what follows.
> >
> > ---
> > **[W1-1] Fine-tuning classifiers with LoRA [1] is not applicable to them?**
> >
> > We acknowledge that LoRA [1] can indeed be independently applied to other baseline methods; our intent was to emphasize that most of the prior works we compared against did not consider classifier fine-tuning, making it challenging to evaluate the specific impact of applying LoRA in the context of certified robustness. We note that the particular effectiveness of LoRA fine-tuning in boosting certified robustness is one of the important findings of our work, e.g., we find that the ACR gain from LoRA nearly matches that of full-parameter tuning (see Table 10).
> >
> >
> > ---
> > **[W1-2]  On the performance of Carlini et al. [2]**
> >
> > Thank you for your comment. We agree with you that Carlini et al. [2] is indeed an important baseline and should not be dismissed for our discussion. Here, we rather wanted to highlight its particular weakness on high radii compared to the prior methods without denoising, which has been a core performance bottleneck of denoising-based methods. We believe one of our key contributions is in identifying and addressing the underlying reason of this issue, i.e., the hallucination effect of the denoiser, achieving improved robustness especially at large radii while maintaining superiority at small radii.
> >
> > Additionally, we clarify a potential confusion that [2] outperforms FT-CADIS at small radii when using the same classifier; for instances, FT-CADIS outperforms [2] with the same ViT model on CIFAR-10 (in Table 1(a)), and even with a smaller model on ImageNet (87M for FT-CADIS and 305M for [2] in Table 1(b)). Although FT-CADIS indeed shows lower clean accuracy in Table 11 that we additionally supplied, this can be usually addressed by simply adjusting the trade-off hyperparameter $\lambda$, while our report for Table 11 was mainly based on higher Averaged Certified Radius (ACR).
> >
> > [1] Hu et al., LoRA: Low-Rank Adaptation of Large Language Models, ICLR 2022 \
> > [2] Carlini et al., (Certified!!) Adversarial Robustness for Free!, ICLR 2023

---

> > > ### Comment · Reviewer_wc3e · 2024-10-14
> > >
> > > Thanks for the response. My point was not about the performance of the proposed method, but rather the discussion about efficiency should be more clear (I think the two first claims in [W1] above are imprecise) and explicit in the text.

---

> > > > ### Author Response · Authors · 2024-10-17
> > > > **Response to Reviewer wc3e**
> > > >
> > > > Dear reviewer wc3e,
> > > >
> > > > Thank you again for your clarification. Following your comment, we have included additional discussion about efficiency in Appendix E of the revised manuscript; we have marked the corresponding revisions with “$\text{\color{blue}blue}$”, for your convenience.
> > > >
> > > > If you have any additional comments or suggestions, please feel free to let us know. We are dedicated to further improving the quality of our work and greatly appreciate your feedback.

---

### Review · Reviewer_AF4i · 2024-09-16

**Summary Of Contributions:**

The authors propose an approach called FT-CADIS training models with certified robustness using denoising models. The approach is based on the denoised smoothing approach which puts a denoiser before an off-the-shelf classifier to deal with the noise injected to input for certified robustness. The authors improve this approach by adding an example filtering process by removing denoised images that are misclassified (what they refer to as hallucinated images), and adding an adversarial loss to the training. Their approach achieves better certified robustness compared to other state-of-art approaches.

**Audience:**

Yes

**Claims And Evidence:**

Yes

**Requested Changes:**

Provide answers to questions listed in weaknesses above.

**Strengths And Weaknesses:**

Strengths:
1. The paper is clearly written and the overall approach is quite easy to understand. The main idea is to filter out hallucinated examples from the denoiser and add an adversarial loss component.
2. The method achieves better certified robustness compared to other state-of-art denoised smoothing approaches
3. There are also good ablation studies analyzing the contributions of each component of the method.


Weaknesses:
1. Conceptually I don't know if filtering out the 'hallucinated' images is a good idea. Those images are still recognizable as the target class by humans, but just misclassified by the off-the-shelf classifier. They represent the distribution shift between the denoised images and clean images. Adapting to these examples could be useful since we are going to use the denoiser during test time and we cannot avoid these distribution shifts. This is analogous to the generation of adversarial examples during adversarial training, we deliberately generate misclassified examples so that the classifier can cope better with adversarial examples during test time when there is an adversary.
Apart from the empirical results do the authors have any explanation why filtering out those 'hallucinated' images help?
2. From Table 3 we can see that under low-noise conditions the certified robustness with 'hallucinated' images are better than the authors' filtering approach. Is this phenomenon related to the inability of the denoiser to denoise the images under high noise conditions?
3. I cannot find the details of LoRA finetuning in the paper.

---

> ### Author Response · Authors · 2024-09-28
> **Response to Reviewer AF4i**
>
> Dear reviewer AF4i,
>
> We sincerely appreciate your efforts in reviewing our manuscript. We respond to each comment in the following content. In the revised manuscript, we have marked the revisions with “red”.
>
> ---
> **[W1] Is filtering out the "hallucinated images" really helpful?**
>
> We first note that there are numerous cases where hallucinated images are so semantically distorted that even humans cannot recognize them as the target class (see Figure 2(a)). Adapting to these "hallucinated" examples can lead to improper learning, as they have lost the semantics of the original assigned class. Our method fundamentally aims to prevent the model from optimizing on such hallucinated images and instead focuses on learning the correct semantics for each class using non-hallucinated images. For your information, we have uploaded additional hallucinated examples in Appendix H of our revised manuscript.
>
> ---
> **[W2] Relationship between the denoiser’s inability at high noise and why “hallucinated images” help improve performance at small radii.**
>
> Thank you for the opportunity to clarify this point. However, there seems to be a misunderstanding regarding Table 3. The noise level is fixed at 1.0, and all certification is conducted under this high-noise condition. We assume that your question regarding the comparison between the first and second rows, where the second row shows higher certified accuracy at epsilon 0.00 and 0.25. We hypothesize that this phenomenon is caused by the limitations of the off-the-shelf classifiers. As the off-the-shelf classifiers are not perfect, i.e., they do not achieve 100\% accuracy, the classifiers may make mistakes in selecting denoised images, which can affect the overall generalizability of the smoothed classifier. We believe this issue can be mitigated with future advanced off-the-shelf classifiers.
>
> ---
> **[W3] No LoRA fine-tuning details.**
>
> We kindly remark that we have provided the details of LoRA fine-tuning in Appendix B.2 (see ImageNet fine-tuning section) and Table 6(b). For your information, we freeze the original model except for the classification layer, and incorporate LoRA weights into the query and value projection matrices of the ViT’s self-attention layers.

---

### Decision · Action_Editor_gm3R · 2024-10-31

**Recommendation:** Accept as is

**Comment:**

As highlighted by all reviewers, this paper proposes a new method to improve certified robustness of image classifiers via denoised smoothing. A problem of interest to the community. The approach is well-motivated, and the experimental results support it, often outperforming the existing methods.

**Audience:**

This paper provides a new technique to improve certified robustness of image classifiers. This is a problem of longstanding interest for the machine learning robustness community that reads TMLR.

**Claims And Evidence:**

The paper provides multiple ablation studies and solid experimental results to back up its claims.